# Sample-Efficient Reinforcement Learning for Linearly-Parameterized MDPs with a Generative Model

**Bingyan Wang**[*]
Princeton University
bingyanw@princeton.edu

**Yuling Yan**[*]
Princeton University
yulingy@princeton.edu

**Jianqing Fan**
Princeton University
jqfan@princeton.edu

## Abstract

The curse of dimensionality is a widely known issue in reinforcement learning (RL). In the tabular setting where the state space $\mathcal{S}$ and the action space $\mathcal{A}$ are both finite, to obtain a nearly optimal policy with sampling access to a generative model, the minimax-optimal sample complexity scales linearly with $|\mathcal{S}| \times |\mathcal{A}|$, which can be prohibitively large when $\mathcal{S}$ or $\mathcal{A}$ is large. This paper considers a Markov decision process (MDP) that admits a set of state-action features, which can linearly express (or approximate) its probability transition kernel. We show that a model-based approach (resp. Q-learning) provably learns an $\varepsilon$-optimal policy (resp. Q-function) with high probability as soon as the sample size exceeds the order of $\frac{K}{(1-\gamma)^3 \varepsilon^2}$ (resp. $\frac{K}{(1-\gamma)^4 \varepsilon^2}$), up to some logarithmic factor. Here $K$ is the feature dimension and $\gamma \in (0, 1)$ is the discount factor of the MDP. The results is applicable to the tabular MDPs by taking the coordinate basis with $K = |\mathcal{S}| \times |\mathcal{A}|$. Both sample complexity bounds are provably tight, and our result for the model-based approach matches the minimax lower bound. Our results show that for arbitrarily large-scale MDP, both the model-based approach and Q-learning are sample-efficient when $K$ is relatively small, and hence the title of this paper.

## 1   Introduction

Reinforcement learning (RL) studies the problem of learning and decision making in a Markov decision process (MDP). Recent years have seen exciting progress in applications of RL in real world decision-making problems such as AlphaGo [46, 47] and autonomous driving [33]. Specifically, the goal of RL is to search for an optimal policy that maximizes the cumulative reward, based on sequential noisy data. There are two popular approaches to RL: model-based and model-free ones.

- The model-based approaches start with formulating an empirical MDP by learning the probability transition model from the collected data samples, and then estimating the optimal policy / value function based on the empirical MDP.

- The model-free approaches (e.g. Q-learning) learn the optimal policy or the optimal (action-)value function from samples. As its name suggests, model-free approaches do not attempt to learn the model explicitly.

---

[*]Equal contribution.

35th Conference on Neural Information Processing Systems (NeurIPS 2021).

Generally speaking, model-based approaches enjoy great flexibility since after the transition model is learned in the first place, it can then be applied to any other problems without touching the raw data samples. In comparison, model-free methods, due to its online nature, are usually memory-efficient and can interact with the environment and update the estimate on the fly.

This paper is devoted to investigating the sample efficiency of both model-based RL and Q-learning (arguably one of the most commonly adopted model-free RL algorithms). It is well known that MDPs suffer from the curse of dimensionality. For example, in the tabular setting where the state space $\mathcal{S}$ and the action space $\mathcal{A}$ are both finite, to obtain a near optimal policy or value function given sampling access to a generative model, the minimax optimal sample complexity scales linearly with $|\mathcal{S}| \times |\mathcal{A}|$ [5, 2]. However contemporary applications of RL often encounters environments with exceedingly large state and action spaces, whilst the data collection might be expensive or even high-stake. This suggests a large gap between the theoretical findings and practical decision-making problems where $|\mathcal{S}|$ and $|\mathcal{A}|$ are large or even infinite.

To close the aforementioned theory-practice gap, one natural idea is to impose certain structural assumption on the MDP. In this paper we follow the feature-based linear transition model studied in [63], where each state-action pair $(s, a) \in \mathcal{S} \times \mathcal{A}$ admits a $K$ dimensional feature vector $\phi(s, a) \in \mathbb{R}^K$ that expresses the transition dynamics $\mathbb{P}(\cdot|s, a) = \Psi \phi(s, a)$ for some unknown matrix $\Psi \in \mathbb{R}^{|\mathcal{S}| \times K}$ which is common for all $(s, a)$. This model encompasses both the tabular case and the homogeneous model in which the state space can be partitioned into $K$ equivalent classes. Assuming access to a generative model [31, 32], under this structural assumption, this paper aims to answer the following two questions:

*How many samples are needed for model-based RL and Q-learning to learn an optimal policy under the feature-based linear transition model?*

In what follows, we will show that the answer to this question scales linearly with the dimension of the feature space $K$ and is independent of $|\mathcal{S}|$ and $|\mathcal{A}|$ under the feature-based linear transition model. With the aid of this structural assumption, model-based RL and Q-learning becomes significantly more sample-efficient than that in the tabular setting.

**Our contributions.** We focus our attention on an infinite horizon MDP with discount factor $\gamma \in (0, 1)$. We use $\varepsilon$-optimal policy to indicate the policy whose expected discounted cumulative rewards are $\varepsilon$ close to the optimal value of the MDP. Our contributions are two-fold:

- We demonstrate that model-based RL provably learns an $\varepsilon$-optimal policy by performing planning based on an empirical MDP constructed from a total number of

$$\widetilde{O}\left(\frac{K}{(1-\gamma)^3 \varepsilon^2}\right)$$

samples, for all $\varepsilon \in (0, (1-\gamma)^{-1/2}]$. Here $\widetilde{O}(\cdot)$ hides logarithmic factors compared to the usual $O(\cdot)$ notation. To the best of our knowledge, this is the first theoretical guarantee for model-based RL under the feature-based linear transition model. This sample complexity bound matches the minimax limit established in [63] up to logarithmic factor.

- We also show that Q-learning provably finds an entrywise $\varepsilon$-optimal Q-function using a total number of

$$\widetilde{O}\left(\frac{K}{(1-\gamma)^4 \varepsilon^2}\right)$$

samples, for all $\varepsilon \in (0, 1]$. This sample complexity upper bound improves the state-of-the-art result in [63] and the dependency on the effective horizon $(1-\gamma)^{-4}$ is sharp in view of [34].

These results taken collectively show the minimax optimality of model-based RL and the sub-optimality of Q-learning in sample complexity.

## 2 Problem formulation

This paper focuses on tabular MDPs in the discounted infinite-horizon setting [10]. Here and throughout, $\Delta_{d-1} := \{v \in \mathbb{R}^d : \sum_{i=1}^{d} v_i = 1, v_i \geq 0, \forall i \in [d]\}$ stands for the $d$-dimensional probability simplex and $[N] := \{1, 2, \cdots, N\}$ for any $N \in \mathbb{N}^+$.

**Discounted infinite-horizon MDPs.** Denote a discounted infinite-horizon MDP by a tuple $M = (\mathcal{S}, \mathcal{A}, P, r, \gamma)$, where $\mathcal{S} = \{1, \cdots, |\mathcal{S}|\}$ is a finite set of states, $\mathcal{A} = \{1, \cdots, |\mathcal{A}|\}$ is a finite set of actions, $P : \mathcal{S} \times \mathcal{A} \to \Delta_{|\mathcal{S}|-1}$ represents the probability transition kernel where $P(s'|s, a)$ denotes the probability of transiting from state $s$ to state $s'$ when action $a$ is taken, $r : \mathcal{S} \times \mathcal{A} \to [0, 1]$ denotes the reward function where $r(s, a)$ is the instantaneous reward received when taking action $a \in \mathcal{A}$ while in state $s \in \mathcal{S}$, and $\gamma \in (0, 1)$ is the discount factor.

**Value function and Q-function.** Recall that the goal of RL is to learn a policy that maximizes the cumulative reward, which corresponds to value functions or Q-functions in the corresponding MDP. For a deterministic policy $\pi : \mathcal{S} \to \mathcal{A}$ and a starting state $s \in \mathcal{S}$, we define the value function as

$$V^\pi(s) := \mathbb{E}\left[\sum_{k=0}^{\infty} \gamma^k r(s_k, a_k) \,\Big|\, s_0 = s\right]$$

for all $s \in \mathcal{S}$. Here, the trajectory is generated by $a_k = \pi(s_k)$ and $s_{k+1} \sim P(s_{k+1}|s_k, a_k)$ for every $k \geq 0$. This function measures the expected discounted cumulative reward received on the trajectory $\{(s_k, a_k)\}_{k \geq 0}$ and the expectation is taken with respect to the randomness of the transitions $s_{k+1} \sim P(\cdot|s_k, a_k)$ on the trajectory. Recall that the immediate rewards lie in $[0, 1]$, it is easy to derive that $0 \leq V^\pi(s) \leq \frac{1}{1-\gamma}$ for any policy $\pi$ and state $s$. Accordingly, we define the Q-function for policy $\pi$ as

$$Q^\pi(s, a) := \mathbb{E}\left[\sum_{k=0}^{\infty} \gamma^k r(s_k, a_k) \,|\, s_0 = s, a_0 = a\right]$$

for all $(s, a) \in \mathcal{S} \times \mathcal{A}$. Here, the actions are chosen by the policy $\pi$ except for the initial state (i.e. $a_k = \pi(s_k)$ for all $k \geq 1$). Similar to the value function, we can easily check that $0 \leq Q^\pi(s, a) \leq \frac{1}{1-\gamma}$ for any $\pi$ and $(s, a)$. To maximize the value function or Q function, previous literature [9, 51] establishes that there exists an optimal policy $\pi^\star$ which simultaneously maximizes $V^\pi(s)$ (resp. $Q^\pi(s, a)$) for all $s \in \mathcal{S}$ (resp. $(s, a) \in \mathcal{S} \times \mathcal{A}$). We define the optimal value function $V^\star$ and optimal Q-function $Q^\star$ respectively as

$$V^\star(s) := \max_\pi V^\pi(s) = V^{\pi^\star}(s), \qquad Q^\star(s, a) := \max_\pi Q^\pi(s, a) = Q^{\pi^\star}(s, a)$$

for any state-action pair $(s, a) \in \mathcal{S} \times \mathcal{A}$.

**Linear transition model.** Given a set of $K$ feature functions $\phi_1, \phi_2, \cdots, \phi_K : \mathcal{S} \times \mathcal{A} \to \mathbb{R}$, we define $\phi$ to be a feature mapping from $\mathcal{S} \times \mathcal{A}$ to $\mathbb{R}^K$ such that

$$\phi(s, a) = [\phi_1(s, a), \cdots, \phi_K(s, a)] \in \mathbb{R}^K.$$

Then we are ready to define the linear transition model [63] as follows.

**Definition 1** (Linear transition model). *Given a discounted infinite-horizon MDP $M = (\mathcal{S}, \mathcal{A}, P, r, \gamma)$ and a feature mapping $\phi : \mathcal{S} \times \mathcal{A} \to \mathbb{R}^K$, $M$ admits the linear transition model if there exists some (unknown) functions $\psi_1, \cdots, \psi_K : \mathcal{S} \to \mathbb{R}$, such that*

$$P(s'|s, a) = \sum_{k=1}^{K} \phi_k(s, a) \psi_k(s') \tag{1}$$

*for every $(s, a) \in \mathcal{S} \times \mathcal{A}$ and $s' \in \mathcal{S}$.*

Readers familiar with linear MDP literatures might immediately recognize that the above definition is the same as the structure imposed on the probability transition kernel $P$ in the linear MDP model [63, 30, 65, 26, 52, 56, 60]. However unlike linear MDP which also requires the reward function $r(s, a)$ to be linear in the feature mapping $\phi(s, a)$, here we do not impose any structural assumption on the reward.

**Example 1** (Tabular MDP). *Each tabular MDP can be viewed as a linear transition model with feature mapping $\phi(s,a) = \boldsymbol{e}_{(s,a)} \in \mathbb{R}^{|\mathcal{S}| \times |\mathcal{A}|}$ (i.e. the vector with all entries equal to 0 but the one corresponding to $(s,a)$ equals to 1) for all $(s,a) \in \mathcal{S} \times \mathcal{A}$. To see this, we can check that Definition 1 is satisfied with $K = |\mathcal{S}| \times |\mathcal{A}|$ and $\psi_{(s,a)}(s') = \mathbb{P}(s'|s,a)$ for each $s, s' \in \mathcal{S}$ and $a \in \mathcal{A}$. This example is a sanity check of Definition 1, which also shows that our results (Theorem 1 and 2) can recover previous results on tabular MDP [2, 34] by taking $K = |\mathcal{S}| \times |\mathcal{A}|$.*

**Example 2** (Simplex Feature Space). *If all feature vectors $\{\phi(s,a)\}_{(s,a) \in \mathcal{S} \times \mathcal{A}}$ fall in the probability simplex $\Delta_{K-1}$, a linear transition model can be constructed by taking $\psi_k(\cdot)$ to be any probability measure over $\mathcal{S}$ for all $k \in [K]$.*

A key observation is that the model size of linear transition model with known feature mapping $\phi$ is $|\mathcal{S}|K$ (the number of coefficients $\psi_k(s')$ in (1)), which is still large when the state space $\mathcal{S}$ is large. In contrast, it will be established later that to learn a near-optimal policy or Q-function, we only need a much smaller number of samples, which depends linearly on $K$ and is independent of $|\mathcal{S}|$.

Next, we introduce a critical assumption employed in prior literature [63, 67, 45].

**Assumption 1** (Anchor state-action pairs). *Assume there exists a set of anchor state-action pairs $\mathcal{K} \subset \mathcal{S} \times \mathcal{A}$ with $|\mathcal{K}| = K$[2] such that for any $(s,a) \in \mathcal{S} \times \mathcal{A}$, its corresponding feature vector can be expressed as a convex combination of the feature vectors of anchor state-action pairs $\{(s,a) : (s,a) \in \mathcal{K}\}$:*

$$\phi(s,a) = \sum_{i:(s_i,a_i) \in \mathcal{K}} \lambda_i(s,a)\,\phi(s_i,a_i) \quad for \quad \sum_{i=1}^{K} \lambda_i(s,a) = 1 \quad and \quad \lambda_i(s,a) \geq 0. \quad (2)$$

*Further, we assume that the vectors in $\{\phi(s,a) : (s,a) \in \mathcal{K}\}$ are linearly independent.*

We pause to develop some intuition of this assumption using Examples 1 and 2. In Example 1, it is straightforward to check that tabular MDPs satisfies Assumption 1 with $\mathcal{K} = \mathcal{S} \times \mathcal{A}$. In terms of Example 2, without loss of generality we can assume that the subspace spanned by the features has full rank, i.e. $\mathsf{span}\{\phi(s,a) : (s,a) \in \mathcal{S} \times \mathcal{A}\} = \mathbb{R}^K$ (otherwise we can reduce the dimension of feature space). Then we can also check that Example 2 satisfies Assumption 1 with arbitrary $\mathcal{K} \subseteq \mathcal{S} \times \mathcal{A}$ such that the vectors in $\{\phi(s,a) : (s,a) \in \mathcal{K}\}$ are linearly independent. In fact, this sort of "anchor" notion appears widely in the literature: [3] considers "anchor word" in topic modeling; [19] defines "separability" in their study of non-negative matrix factorization; [48] introduces "aggregate" in reinforcement learning; [21] studies "anchor state" in soft state aggregation models. These concepts all bear some kind of resemblance to our definition of anchor state-action pairs here.

Throughout this paper, we assume that the feature mapping $\phi$ is known, which is a widely adopted assumption in previous literature [63, 30, 68, 26, 52, 56, 60]. In practice, large scale RL usually makes use of representation learning to obtain the feature mapping $\phi$. Furthermore, the learned representations can be selected to satisfy the anchor state-action pairs assumption by design.

A useful implication of Assumption 1 is that we can represent the transition kernel as

$$P(\cdot|s,a) = \sum_{i:(s_i,a_i) \in \mathcal{K}} \lambda_i(s,a)\,P(\cdot|s_i,a_i), \quad (3)$$

This follows simply from substituting (2) into (1) (see (14) in Appendix A for a formal proof).

## 3 Model-based RL with a generative model

We start with studying model-based RL with a generative model in this section. We propose a model-based planning algorithm and show that it returns an $\varepsilon$-optimal policy with minimax optimal sample size.

### 3.1 Main results

**A generative model and an empirical MDP.** We assume access to a generative model that provides us with independent samples from $M$. For each anchor state-action pair $(s_i, a_i) \in \mathcal{K}$, we collect $N$

---

[2]Without loss of generality, one can always assume that the number of anchor state-action pairs equals to the feature dimension $K$. Interested readers are referred to Appendix D for detailed argument.

independent samples $s_i^{(j)} \sim P(\cdot | s_i, a_i)$, $j \in [N]$. This allows us to construct an empirical transition kernel $\widehat{P}$ where

$$\widehat{P}(s' \mid s, a) = \sum_{i=1}^{K} \lambda_i(s, a) \cdot \left( \frac{1}{N} \sum_{j=1}^{N} \mathbb{1}\left\{ s_i^{(j)} = s' \right\} \right), \tag{4}$$

for each $(s, a) \in \mathcal{S} \times \mathcal{A}$. Here, $\frac{1}{N} \sum_{j=1}^{N} \mathbb{1}\{s_i^{(j)} = s'\}$ is an empirical estimate of $P(s'|s_i, a_i)$ and then (3) is employed. With $\widehat{P}$ in hand, we can construct an empirical MDP $\widehat{M} = (\mathcal{S}, \mathcal{A}, \widehat{P}, r, \gamma)$. Our goal here is to derive the sample complexity which guarantees that the optimal policy of $\widehat{M}$ is an $\varepsilon$-optimal policy for the true MDP $M$. The algorithm is summarized below.

---

**Algorithm 1** Model-based RL with a generative model

---

**Inputs**: a set of anchor state-action pairs $\mathcal{K} = \{(s_i, a_i) : i \in [K]\}$, feature mapping $\phi : \mathcal{S} \times \mathcal{A} \to \mathbb{R}^K$, any planning algorithm $\mathcal{P}$, target algorithmic error level $\varepsilon_{\text{opt}}$.
**For** $i = 1, \cdots, K$ **do**
    Draw $N$ independent samples $s_i^{(j)} \sim P(\cdot | s_i, a_i)$, $j = 1, \cdots, N$.
**End for**
Construct an empirical MDP $\widehat{M} = (\mathcal{S}, \mathcal{A}, \widehat{P}, r, \gamma)$ where $\widehat{P}$ can be computed by (4).
**Output** $\widehat{\pi}$ as an $\varepsilon_{\text{opt}}$-optimal policy of $\widehat{M}$ computed by the planning algorithm $\mathcal{P}$.

---

Careful readers may note that in Algorithm 1, $\{\lambda(s, a) : (s, a) \in \mathcal{S} \times \mathcal{A}\}$ is used in the construction of $\widehat{P}$, while $\{\lambda(s, a) : (s, a) \in \mathcal{S} \times \mathcal{A}\}$ is not input into the algorithm. This is because given $\mathcal{K}$ and $\phi$ are known, $\{\lambda(s, a) : (s, a) \in \mathcal{S} \times \mathcal{A}\}$ can be calculated explicitly. The following theorem provides theoretical guarantees for the output policy $\widehat{\pi}$ of the chosen optimization algorithm on the empirical MDP $\widehat{M}$.

**Theorem 1.** *Suppose that $\delta > 0$ and $\varepsilon \in (0, (1-\gamma)^{-1/2}]$. Let $\widehat{\pi}$ be the policy returned by Algorithm 1. Assume that*

$$N \geq \frac{C \log \left( K / \left( (1 - \gamma) \delta \right) \right)}{(1 - \gamma)^3 \varepsilon^2} \tag{5}$$

*for some sufficiently large constant $C > 0$. Then with probability exceeding $1 - \delta$,*

$$Q^\star(s, a) - Q^{\widehat{\pi}}(s, a) \leq \varepsilon + \frac{4\varepsilon_{\text{opt}}}{1 - \gamma}, \tag{6}$$

*for every $(s, a) \in \mathcal{S} \times \mathcal{A}$. Here $\varepsilon_{\text{opt}}$ is the target algorithmic error level in Algorithm 1.*

We first remark that the two terms on the right hand side of (6) can be viewed as statistical error and algorithmic error, respectively. The first term $\varepsilon$ denotes the statistical error coming from the deviation of the empirical MDP $\widehat{M}$ from the true MDP $M$. As the sample size $N$ grows, $\varepsilon$ could decrease towards 0. The other term $4\varepsilon_{\text{opt}}/(1 - \gamma)$ represents the algorithmic error where $\varepsilon_{\text{opt}}$ is the target accuracy level of the planning algorithm applied to $\widehat{M}$. Note that $\varepsilon_{\text{opt}}$ can be arbitrarily small if we run the planning algorithm (e.g. value iteration) for enough iterations. A few implications of this theorem are in order.

- *Minimax-optimal sample complexity.* Assume that $\varepsilon_{\text{opt}}$ is made negligibly small, e.g. $\varepsilon_{\text{opt}} = O((1 - \gamma)\varepsilon)$ to be discussed in the next point. Note that we draw $N$ independent samples for each state-action pair $(s, a) \in \mathcal{K}$, therefore the requirement (5) for finding an $O(\varepsilon)$-optimal policy translates into the following sample complexity requirement

$$\widetilde{O} \left( \frac{K}{(1 - \gamma)^3 \varepsilon^2} \right).$$

This matches the minimax optimal lower bound (up to a logarithm factor) established in [63, Theorem 1] for feature-based MDP. In comparison, for tabular MDP the minimax optimal sample complexity is $\widetilde{\Omega}((1-\gamma)^{-3}\varepsilon^{-2}|\mathcal{S}||\mathcal{A}|)$ [5, 2]. Our sample complexity scales linearly with $K$ instead of $|\mathcal{S}||\mathcal{A}|$ for tabular MDP as desired.

- *Computational complexity.* An advantage of Theorem 1 is that it incorporates the use of any efficient planning algorithm applied to the empirical MDP $\widehat{M}$. Classical algorithms include Q-value iteration (QVI) or policy iteration (PI) [43]. For example, QVI achieves the target level $\varepsilon_{\mathsf{opt}}$ in $O((1-\gamma)^{-1}\log\varepsilon_{\mathsf{opt}}^{-1})$ iterations, and each iteration takes time proportional to $O(NK + |\mathcal{S}||\mathcal{A}|K)$. To learn an $O(\varepsilon)$-optimal policy, which requires sample complexity (5) and the target level $\varepsilon_{\mathsf{opt}} = O((1-\gamma)\varepsilon)$, the overall running time is

$$\widetilde{O}\left(\frac{|\mathcal{S}|\,|\mathcal{A}|\,K}{1-\gamma} + \frac{K}{(1-\gamma)^4\,\varepsilon^2}\right).$$

  In comparison, for the tabular MDP the corresponding running time is $\widetilde{O}((1-\gamma)^{-4}\varepsilon^{-2}|\mathcal{S}||\mathcal{A}|)$ [2]. This suggests that under the feature-based linear transition model, the computational complexity is $\min\{|\mathcal{S}||\mathcal{A}|/K, (1-\gamma)^{-3}\varepsilon^{-2}/K\}$ times lower than that for the tabular MDP (up to logarithm factors), which is significantly more efficient when $K$ is not too large.

- *Stability vis-à-vis model misspecification.* A more general version of Theorem 1 (Theorem 3 in Appendix B) shows that when $P$ approximately (instead of exactly) admits the linear transition model, we can still achieve some meaningful result. Specifically, if there exists a linear transition kernel $\widetilde{P}$ obeying $\max_{(s,a)\in\mathcal{S}\times\mathcal{A}}\|\widetilde{P}(\cdot|s,a) - P(\cdot|s,a)\|_1 \le \xi$ for some $\xi \ge 0$, we can show that $\widehat{\pi}$ returned by Algorithm 1 (with slight modification) satisfies

$$Q^\star(s,a) - Q^{\widehat{\pi}}(s,a) \le \varepsilon + \frac{4\varepsilon_{\mathsf{opt}}}{1-\gamma} + \frac{22\xi}{(1-\gamma)^2},$$

  for every $(s,a) \in \mathcal{S} \times \mathcal{A}$. This shows that the model-based method is stable vis-a-vis model misspecification. Interested readers are referred to Appendix B for more details.

In Algorithm 1, the reward function $r$ is assumed to be known. If the information of $r$ is unavailable, an alternative is to assume that $r$ is linear with respect to the feature mapping $\phi$, i.e. $r(s,a) = \theta^\top \phi(s,a)$ for every $(s,a) \in \mathcal{S} \times \mathcal{A}$, which is widely adopted in linear MDP literature [26, 30, 56, 60]. Under this linear assumption, one can obtain $\theta$ by solving the following linear system of equations

$$r(s,a) = \theta^\top \phi(s,a), \quad \forall (s,a) \in \mathcal{K}, \tag{7}$$

which can be constructed by the observed reward $r(s,a)$ for all anchor state-action pairs.

## 4 Model-free RL—vanilla Q Learning

In this section, we turn to study one of the most popular model-free RL algorithms—Q-learning. We provide tight sample complexity bound for vanilla Q-learning under the feature-based linear transition model, which shows its sample-efficiency (depends on $|K|$ instead of $|\mathcal{S}|$ or $|\mathcal{A}|$) and sub-optimality in the dependency on the effective horizon.

### 4.1 Q-learning algorithm

The vanilla Q-learning algorithm maintains a Q-function estimate $Q_t : \mathcal{S} \times \mathcal{A} \to \mathbb{R}$ for all $t \ge 0$, with initialization $Q_0$ obeying $0 \le Q_0(s,a) \le \frac{1}{1-\gamma}$ for every $(s,a) \in \mathcal{S} \times \mathcal{A}$. Assume we have access to a generative model. In each iteration $t \ge 1$, we collect an independent sample $s_t(s,a) \sim P(\cdot|s,a)$ for every anchor state-action pair $(s,a) \in \mathcal{K}$ and define $Q_{\mathcal{K}}^{(t)} : \mathcal{K} \to \mathbb{R}$ to be

$$Q_{\mathcal{K}}^{(t)}(s,a) \coloneqq \max_{a'\in\mathcal{A}} Q_t(s_t, a'), \qquad s_t \equiv s_t(s,a) \sim P(\cdot|s,a).$$

Then given the learning rate $\eta_t \in (0,1]$, the algorithm adopts the following update rule to update all entries of the Q-function estimate

$$Q_t = (1 - \eta_t) Q_{t-1} + \eta_t \mathcal{T}_{\mathcal{K}}^{(t)}(Q_{t-1}).$$

Here, $\mathcal{T}_{\mathcal{K}}^{(t)}$ is an empirical Bellman operator associated with the linear transition model $M$ and the set $\mathcal{K}$ and is given by

$$\mathcal{T}_{\mathcal{K}}^{(t)}(Q)(s,a) \coloneqq r(s,a) + \gamma\lambda(s,a) Q_{\mathcal{K}}^{(t)},$$

where (3) is used in the construction. Clearly, this newly defined operator $\mathcal{T}_{\mathcal{K}}^{(t)}$ is an unbiased estimate of the famous Bellman operator $\mathcal{T}$ [8] defined as

$$\forall (s,a) \in \mathcal{S} \times \mathcal{A}: \qquad \mathcal{T}(Q)(s,a) \coloneqq r(s,a) + \gamma \mathbb{E}_{s' \sim P(\cdot|s,a)} \left[ \max_{a' \in \mathcal{A}} Q(s',a') \right].$$

A critical property is that the Bellman operator $\mathcal{T}$ is contractive with a unique fixed point which is the optimal Q-function $Q^\star$[8]. To solve the fixed-point equation $\mathcal{T}(Q^\star) = Q^\star$, Q-learning was then introduced by [58] based on the idea of stochastic approximation [44]. This procedure is precisely described in Algorithm 2.

---

**Algorithm 2** Vanilla Q-learning for infinite-horizon discounted MDPs

---

**inputs**: learning rates $\{\eta_t\}$, number of iterations $T$, discount factor $\gamma$, initial estimate $Q_0$.
**for** $t = 1, \ldots, T$ **do**
    Draw $s_t(s,a) \sim P(\cdot|s,a)$ for each $(s,a) \in \mathcal{K}$.
    Compute $\boldsymbol{Q}_t$ according to the update rule
$$Q_t = (1 - \eta_t) Q_{t-1} + \eta_t \mathcal{T}_{\mathcal{K}}^{(t)}(Q_{t-1}).$$
**end for**

---

## 4.2 Main results

We are now ready to provide our main result for vanilla Q-learning, assuming sampling access to a generative model.

**Theorem 2.** *Consider any $\delta \in (0,1)$ and $\varepsilon \in (0,1]$. Assume that for any $0 \leq t \leq T$, the learning rates satisfy*

$$\frac{1}{1 + \frac{c_1(1-\gamma)T}{\log^2 T}} \leq \eta_t \leq \frac{1}{1 + \frac{c_2(1-\gamma)t}{\log^2 T}} \tag{8}$$

*for some sufficiently small universal constants $c_1 \geq c_2 > 0$. Suppose that the total number of iterations $T$ exceeds*

$$T \geq \frac{C_3 \log(KT/\delta) \log^4 T}{(1-\gamma)^4 \varepsilon^2} \tag{9}$$

*for some sufficiently large universal constant $C_3 > 0$. If the initialization obeys $0 \leq Q_0(s,a) \leq \frac{1}{1-\gamma}$ for any $(s,a) \in \mathcal{S} \times \mathcal{A}$, then with probability exceeding $1 - \delta$, the output $Q_T$ of Algorithm 2 satisfies*

$$\max_{(s,a) \in \mathcal{S} \times \mathcal{A}} |Q_T(s,a) - Q^\star(s,a)| \leq \varepsilon. \tag{10}$$

*In addition, let $\pi_T$ (resp. $V_T$) to be the policy (resp. value function) induced by $Q_T$, then one has*

$$\max_{s \in \mathcal{S}} |V^{\pi_T}(s) - V^\star(s)| \leq \frac{2\gamma\varepsilon}{1-\gamma}. \tag{11}$$

This theorem provides theoretical guarantees on the performance of Algorithm 2. A few implications of this theorem are in order.

- *Learning rate*. The condition (8) accommodates two commonly adopted choice of learning rates: (i) linearly rescaled learning rates $\eta_t = [1 + c_2(1-\gamma)t/\log^2 T]^{-1}$, and (ii) iteration-invariant learning rates $\eta_t \equiv [1 + c_1(1-\gamma)T/\log^2 T]$. Interested readers are referred to the discussions in [34, Section 3.1] for more details on these two learning rate schemes.

- *Tight sample complexity bound*. Note that we draw $K$ independent samples in each iteration, therefore the iteration complexity (9) can be translated into the sample complexity bound $TK$ in order for Q-learning to achieve $\varepsilon$-accuracy:

$$\widetilde{O}\left(\frac{K}{(1-\gamma)^4 \varepsilon^2}\right). \tag{12}$$

As we will see shortly, this result improves the state-of-the-art sample complexity bound presented in [63, Theorem 2] . In addition, the dependency on the effective horizon $(1-\gamma)^{-4}$ matches the lower bound established in [34, Theorem 2] for vanilla Q-learning using either learning rate scheme covered in the previous remark, suggesting that our sample complexity bound (12) is sharp.

- *Stability vis-à-vis model misspecification.* Just like the model-based approach, we can also show that Q-learning is also stable vis-a-vis model misspecification when $P$ approximately admits the linear transition model. We refer interested readers to Theorem 4 in Appendix B for more details.

**Comparison with [63].** We compare our result with the sample complexity bounds for Q-learning under the feature-based linear transition model in [63].

- We first compare our result with [63, Theorem 2], which is, to the best of our knowledge, the state-of-the-art theory for this problem. When there is no model misspecification, [63, Theorem 2] showed that in order for their Phased Parametric Q-learning[3] (Algorithm 1 therein) to learn an $\varepsilon$-optimal policy, the sample size needs to be

$$\widetilde{O}\left(\frac{K}{(1-\gamma)^7 \varepsilon^2}\right).$$

  Note that (12) is the sample complexity required for entrywise $\varepsilon$-accurate estimate of the optimal Q-function, thus a fair comparison requires to use the sample complexity for learning an $\varepsilon$-optimal policy deduced from (11), which is

$$\widetilde{O}\left(\frac{K}{(1-\gamma)^6 \varepsilon^2}\right).$$

  Hence, our sample complexity improves upon previous work by a factor at least on the order of $(1-\gamma)^{-1}$. However it is worth mentioning that [63, Theorem 2] is built upon weaker conditions $\sum_{i=1}^{K} \lambda_i(s,a) = 1$ and $\sum_{i=1}^{K} |\lambda_i(s,a)| \leq L$ for some $L \geq 1$, which does not require $\lambda_i(s,a) \geq 0$. Our result holds under Assumption 1, which requires $\sum_{i=1}^{K} \lambda_i(s,a) = 1$ and $\lambda_i(s,a) \geq 0$. Under the current analysis framework, it is difficult to obtain tight sample complexity bounds without assuming $\lambda_i(s,a) \geq 0$.

- Besides vanilla Q-learning, [63] also proposed a new variant of Q-learning called Optimal Phased Parametric Q-Learning (Algorithm 2 therein), which is essentially Q-learning with variance reduction. [63, Theorem 3] showed that the sample complexity for this algorithm is

$$\widetilde{O}\left(\frac{K}{(1-\gamma)^3 \varepsilon^2}\right),$$

  which matches minimax optimal lower bound (up to a logarithm factor) established in [63, Theorem 1]. Careful reader might notice that this sample complexity bound is better than ours for vanilla Q-learning. We emphasize that as elucidated in the second implication under Theorem 2, our result is already tight for vanilla Q-learning. This observation reveals that while the sample complexity for vanilla Q-learning is provably sub-optimal, the variants of Q-learning can have better performance and achieve minimax optimal sample complexity.

We conclude this section by comparing model-based and model-free approaches. Theorem 1 shows that the sample complexity of the model-based approach is minimax optimal, whilst vanilla Q-learning, perhaps the most commonly adopted model-free method, is sub-optimal according to Theorem 2. However this does not mean that model-based method is better than model-free ones since (i) some variants of Q-learning (see [63, Algorithm 2] for example) also has minimax optimal sample complexity; and (ii) in many applications it might be unrealistic to estimate the model in advance.

## 5 A glimpse of our technical approaches

The establishment of Theorems 1 and 2 calls for a series of technical novelties in the proof. In what follows, we briefly highlight our key technical ideas and novelties.

---

[3]The difference between Algorithm 2 and Phased Parametric Q-Learning in [63] is that Algorithm 2 maintains and updates a $Q$-function estimate $Q_t$, while Phased Parametric Q-Learning parameterized $Q$-function by

$$Q_w(s,a) := r(s,a) + \gamma \phi(s,a)^\top w,$$

and then updates the parameters $w$.

- For the model-based approach, we employ "leave-one-out" analysis to decouple the complicated statistical dependency between the empirical probability transition model $\widehat{P}$ and the corresponding optimal policy. Specifically, [2] proposed to construct a collection of auxiliary MDPs where each one of them leaves out a single state $s$ by setting $s$ to be an absorbing state and keeping everything else untouched. We tailor this high level idea to the needs of linear transition model, then the independence between the newly constructed MDP with absorbing state $s$ and data samples collected at state $s$ will facilitate our analysis, as detailed in Lemma 1. Compared with [2], Theorem 1 extends the tabular setting studied in [2] to the linear transition model and accommodates model misspecification, which actually needs significant efforts as detailed in the supplementary materials. This "leave-one-out" type of analysis has been utilized in studying numerous problems by a long line of work, such as [22, 38, 53, 13, 12, 14, 15], just to name a few.

- To obtain tighter sample complexity bound than the previous one $\widetilde{O}(\frac{K}{(1-\gamma)^7\varepsilon^2})$ in [63] for vanilla Q-learning, we invoke Freedman's inequality [24] for the concentration of an error term with martingale structure as illustrated in Appendix C, while the classical ones used in analyzing Q-learning are Hoeffding's inequality and Bernstein's inequality [63]. The use of Freedman's inequality helps us establish a recursive relation on $\{\|Q_t - Q^\star\|_\infty\}_{t=0}^T$, which consequently leads to the performance guarantee (10). It is worth mentioning that [34] also studied vanilla Q-learning in the tabular MDP setting and adopted Freedman's inequality, while we emphasize that it requires a lot of efforts and more delicate analyses in order to study linear transition model and also allow for model misspecification in the current paper, as detailed in the supplementary material.

# 6 Additional related literature

To remedy the issue of prohibitively high sample complexity, there exists a substantial body of literature proposing and studying many structural assumptions and complexity notions under different settings. This current paper focuses on linear transition model which is studied in MDP by numerous previous works [63, 30, 64, 68, 40, 25, 56, 52, 26, 60]. Among them, [63] studied linear transition model and provided tight sample complexity bounds for a new variant of Q-learning with the help of variance reduction. [30] focused on linear MDP and designed an algorithm called "Least-Squares Value Iteration with UCB" with both polynomial runtime and polynomial sample complexity without accessing generative model. [56] extended the study of linear MDP to the framework of reward-free reinforcement learning. [68] considered a different feature mapping called linear kernel MDP and devised an algorithm with polynomial regret bound without generative model. Other popular structure assumptions include: [61] studied fully deterministic transition dynamics; [28] introduced Bellman rank and proposed an algorithm which needs sample size polynomially dependent on Bellman rank to obtain a near-optimal policy in contextual decision processes; [20] assumed that the value function has low variance compared to the mean for all deterministic policy; [39, 42, 7, 66] used linear model to approximate the value function; [35] assumed that the optimal Q-function can be linearly-parameterized by the features.

Apart from the linear transition model, another notion adopted in this work is the generative model, whose role in discounted MDP has been studied by extensive literature. The concept of generative model was originally introduced by [32], and then widely adopted in numerous works, including [31, 5, 63, 54, 2, 41], to name a few. Specifically, it is assumed that a generative model of the studied MDP is available and can be queried for every state-action pair and output the next state. Among previous works, [5] proved that the minimax lower bound on the sample complexity to obtain an $\varepsilon$-optimal policy was $\widetilde{\Omega}(\frac{|\mathcal{S}||\mathcal{A}|}{(1-\gamma)^3\varepsilon^2})$. [5] also showed that model-based approach can output an $\varepsilon$-optimal value function with near-optimal sample complexity for $\varepsilon \in (0,1)$. Then [2] made significant progress on the challenging problem of establishing minimax optimal sample complexity in estimating an $\varepsilon$-optimal policy with the help of "leave-one-out" analysis.

In addition, after being proposed in [59], Q-learning has become the focus of a rich line of research [58, 11, 32, 23, 4, 29, 53, 16, 37, 62]. Among them, [16, 37, 62] studied Q-learning in the presence of Markovian data, i.e. a single sample trajectory. In contrast, under the generative setting of Q-learning where a fresh sample can be drawn from the simulator at each iteration, [54] analyzed a variant of Q-learning with the help of variance reduction, which was proved to enjoy minimax optimal sample complexity $\widetilde{O}(\frac{|\mathcal{S}||\mathcal{A}|}{(1-\gamma)^3\varepsilon^2})$. Then more recently, [34] improved the lower bound of the

vanilla Q-learning algorithm in terms of its scaling with $\frac{1}{1-\gamma}$ and proved a matching upper bound $\widetilde{O}(\frac{|\mathcal{S}||\mathcal{A}|}{(1-\gamma)^4\varepsilon^2})$.

## 7 Discussion

This paper studies sample complexity of both model-based and model-free RL under a discounted infinite-horizon MDP with feature-based linear transition model. We establish tight sample complexity bounds for both model-based approaches and Q-learning, which scale linearly with the feature dimension $K$ instead of $|\mathcal{S}| \times |\mathcal{A}|$, thus considerably reduce the required sample size for large-scale MDPs when $K$ is relatively small. Our results are sharp, and the sample complexity bound for the model-based approach matches the minimax lower bound. The current work suggests a couple of directions for future investigation, as discussed in detail below.

- *Extension to episodic MDPs.* An interesting direction for future research is to study linear transition model in episodic MDP. This focus of this work is infinite-horizon discounted MDPs, and hopefully the analysis here can be extended to study the episodic MDP as well ([17, 18, 6, 27, 55, 26]).

- *Continuous state and action space.* The state and action spaces in this current paper are still assumed to be finite, since the proof relies heavily on the matrix operations. However, we expect that the results can be extended to accommodate continuous state and action space by employing more complicated analysis.

- *Accommodating entire range of $\varepsilon$.* Since both value functions and Q-functions can take value in $[0, (1-\gamma)^{-1}]$, ideally our theory should cover all choices of $\varepsilon \in (0, (1-\gamma)^{-1}]$. However we require that $\varepsilon \in (0, (1-\gamma)^{-1/2}]$ in Theorem 1 and $\varepsilon \in (0, 1]$ in Theorem 2. While most of the prior works like [2, 63] also impose these restrictions, a recent work [36] proposed a perturbed model-based planning algorithm and proved minimax optimal guarantees for any $\varepsilon \in (0, (1-\gamma)^{-1}]$. While their work only focused on model-based RL under tabular MDP, an interesting future direction is to improve our theory to accommodate any $\varepsilon \in (0, (1-\gamma)^{-1}]$.

- *General function approximation.* Another future direction is to extend the study to more general function approximation starting from linear structure covered in this paper. There exists a rich body of work proposing and studying different structures, such as linear value function approximation [39, 42, 7, 66], linear MDPs with infinite dimensional features [1], Eluder dimension [57], Bellman rank [28] and Witness rank [50], etc. Therefore, it is hopeful to investigate these settings and improve the sample efficiency.

## Acknowledgments and Disclosure of Funding

B. Wang is supported in part by Gordon Y. S. Wu Fellowships in Engineering. Y. Yan is supported in part by ARO grant W911NF-20-1-0097 and NSF grant CCF-1907661. Part of this work was done while Y. Yan was visiting the Simons Institute for the Theory of Computing. J. Fan is supported in part by the ONR grant N00014-19-1-2120 and the NSF grants DMS-1662139, DMS-1712591, DMS-2052926, DMS-2053832, and the NIH grant 2R01-GM072611-15.

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
