# Supplement to "Sample-Efficient Reinforcement Learning for Linearly-Parameterized MDPs with a Generative Model"

**Bingyan Wang**[*]
Princeton University
bingyanw@princeton.edu

**Yuling Yan**[*]
Princeton University
yulingy@princeton.edu

**Jianqing Fan**
Princeton University
jqfan@princeton.edu

## A    Notations

In this section we gather the notations that will be used throughout the appendix.

For any vectors $\boldsymbol{u} = [u_i]_{i=1}^n \in \mathbb{R}^n$ and $\boldsymbol{v} = [u_i]_{i=1}^n \in \mathbb{R}^n$, let $\boldsymbol{u} \circ \boldsymbol{v} = [u_i v_i]_{i=1}^n$ denote the Hadamard product of $\boldsymbol{u}$ and $\boldsymbol{v}$. We slightly abuse notations to use $\sqrt{\cdot}$ and $|\cdot|$ to define entry-wise operation, i.e. for any vector $\boldsymbol{v} = [v_i]_{i=1}^n$ denote $\sqrt{\boldsymbol{v}} := [\sqrt{v_i}]_{i=1}^n$ and $|\boldsymbol{v}| := [|v_i|]_{i=1}^n$. Furthermore, the binary notations $\leq$ and $\geq$ are both defined in entry-wise manner, i.e. $\boldsymbol{u} \leq \boldsymbol{v}$ (resp. $\boldsymbol{u} \geq \boldsymbol{v}$) means $u_i \leq v_i$ (resp. $u_i \geq v_i$) for all $1 \leq i \leq n$. For a collection of vectors $\boldsymbol{v}_1, \cdots, \boldsymbol{v}_m \in \mathbb{R}^n$ with $\boldsymbol{v}_i = [v_{i,j}]_{j=1}^n \in \mathbb{R}^n$, we define the max operator to be $\max_{1 \leq i \leq m} \boldsymbol{v}_i := [\max_{1 \leq i \leq m} v_{i,j}]_{j=1}^n$.

For any matrix $\boldsymbol{M} \in \mathbb{R}^{m \times n}$, $\|\boldsymbol{M}\|_1$ is defined as the largest row-wise $\ell_1$ norm of $\boldsymbol{M}$, i.e. $\|\boldsymbol{M}\|_1 := \max_i \sum_j |M_{i,j}|$. In addition, we define $\mathbf{1}$ to be a vector with all the entries being 1, and $\boldsymbol{I}$ be the identity matrix. To express the probability transition function $P$ in matrix form, we define the matrix $\boldsymbol{P} \in \mathbb{R}^{|\mathcal{S}||\mathcal{A}| \times |\mathcal{S}|}$ to be a matrix whose $(s,a)$-th row $\boldsymbol{P}_{s,a}$ corresponds to $P(\cdot|s,a)$. In addition, we define $\boldsymbol{P}^\pi$ to be the probability transition matrix induced by policy $\pi$, i.e. $\boldsymbol{P}^\pi_{(s,a),(s',a')} = \boldsymbol{P}_{s,a}(s') \mathbb{1}_{\pi(s')=a'}$ for all state-action pairs $(s,a)$ and $(s',a')$. We define $\pi_t$ to be the policy induced by $Q_t$, i.e. $Q_t(s, \pi_t(s)) = \max_a Q_t(s,a)$ for all $s \in \mathcal{S}$. Furthermore, we denote the reward function $r$ by vector $\boldsymbol{r} \in \mathbb{R}^{|\mathcal{S}||\mathcal{A}|}$, i.e. the $(s,a)$-th element of $\boldsymbol{r}$ equals $r(s,a)$. In the same manner, we define $\boldsymbol{V}^\pi \in \mathbb{R}^{|\mathcal{S}|}, \boldsymbol{V}^\star \in \mathbb{R}^{|\mathcal{S}|}, \boldsymbol{V}_t \in \mathbb{R}^{|\mathcal{S}|}, \boldsymbol{Q}^\pi \in \mathbb{R}^{|\mathcal{S}||\mathcal{A}|}, \boldsymbol{Q}^\star \in \mathbb{R}^{|\mathcal{S}||\mathcal{A}|}$ and $\boldsymbol{Q}_t \in \mathbb{R}^{|\mathcal{S}||\mathcal{A}|}$ to represent $V^\pi, V^\star, V_t, Q^\pi, Q^\star$ and $Q_t$ respectively. By using these notations, we can rewrite the Bellman equation as

$$\boldsymbol{Q}^\pi = \boldsymbol{r} + \gamma \boldsymbol{P} \boldsymbol{V}^\pi = \boldsymbol{r} + \gamma \boldsymbol{P}^\pi \boldsymbol{Q}^\pi. \tag{11}$$

Further, for any vector $\boldsymbol{V} \in \mathbb{R}^{|\mathcal{S}|}$, let $\mathsf{Var}_{\boldsymbol{P}}(\boldsymbol{V}) \in \mathbb{R}^{|\mathcal{S}||\mathcal{A}|}$ be

$$\mathsf{Var}_{\boldsymbol{P}}(\boldsymbol{V}) := \boldsymbol{P}(\boldsymbol{V} \circ \boldsymbol{V}) - (\boldsymbol{P}\boldsymbol{V}) \circ (\boldsymbol{P}\boldsymbol{V}), \tag{12}$$

and define $\mathsf{Var}_{\boldsymbol{P}_{s,a}}(\boldsymbol{V}) \in \mathbb{R}$ to be

$$\mathsf{Var}_{\boldsymbol{P}_{s,a}}(\boldsymbol{V}) := \boldsymbol{P}_{s,a}(\boldsymbol{V} \circ \boldsymbol{V}) - (\boldsymbol{P}_{s,a}\boldsymbol{V})^2, \tag{13}$$

where $\boldsymbol{P}_{s,a}$ is the $(s,a)$-th row of $\boldsymbol{P}$.

Next, we reconsider Assumption 1. For any state-action pair $(s,a)$, we define vector $\boldsymbol{\lambda}(s,a) \in \mathbb{R}^K$ (resp. $\boldsymbol{\phi}(s,a) \in \mathbb{R}^K$) with $\boldsymbol{\lambda}(s,a) = [\lambda_i(s,a)]_{i=1}^K$ (resp. $\boldsymbol{\phi}(s,a) = [\phi_i(s,a)]_{i=1}^K$) and matrix

---

[*]Equal contribution.

35th Conference on Neural Information Processing Systems (NeurIPS 2021).

$\boldsymbol{\Lambda} \in \mathbb{R}^{|\mathcal{S}||\mathcal{A}| \times K}$ (resp. $\boldsymbol{\Phi} \in \mathbb{R}^{|\mathcal{S}||\mathcal{A}| \times K}$) whose $(s,a)$-th row corresponds to $\boldsymbol{\lambda}(s,a)^\top$ (resp. $\boldsymbol{\phi}(s,a)^\top$). Define vector $\boldsymbol{\psi}(s,a) \in \mathbb{R}^K$ with $\boldsymbol{\psi}(s,a) = [\psi_i(s,a)]_{i=1}^K$ and matrix $\boldsymbol{\Psi} \in \mathbb{R}^{K \times |\mathcal{S}|}$ whose $(s,a)$-th column corresponds to $\boldsymbol{\psi}(s,a)^\top$. Further, let $\boldsymbol{P}_\mathcal{K} \in \mathbb{R}^{K \times |\mathcal{S}|}$ (resp. $\boldsymbol{\Phi}_\mathcal{K} \in \mathbb{R}^{K \times K}$) to be a submatrix of $\boldsymbol{P}$ (resp. $\boldsymbol{\Phi}$) formed by concatenating the rows $\{\boldsymbol{P}_{s,a}, (s,a) \in \mathcal{K}\}$ (resp. $\{\boldsymbol{\Phi}_{s,a}, (s,a) \in \mathcal{K}\}$). By using the previous notations, we can express the relations in Definition 1 and Assumption 1 as $\boldsymbol{P}_\mathcal{K} = \boldsymbol{\Phi}_\mathcal{K} \boldsymbol{\Psi}$, $\boldsymbol{P} = \boldsymbol{\Phi} \boldsymbol{\Psi}$ and $\boldsymbol{\Phi} = \boldsymbol{\Lambda} \boldsymbol{\Phi}_\mathcal{K}$. Note that Assumption 1 suggests $\boldsymbol{\Phi}_\mathcal{K}$ is invertible. Taking these equations collectively yields

$$\boldsymbol{P} = \boldsymbol{\Phi}\boldsymbol{\Psi} = \boldsymbol{\Phi}\boldsymbol{\Phi}_\mathcal{K}^{-1}\boldsymbol{P}_\mathcal{K} = \boldsymbol{\Lambda}\boldsymbol{\Phi}_\mathcal{K}\boldsymbol{\Phi}_\mathcal{K}^{-1}\boldsymbol{P}_\mathcal{K} = \boldsymbol{\Lambda}\boldsymbol{P}_\mathcal{K}, \tag{14}$$

which is reminiscent of the anchor word condition in topic modelling [2]. In addition, for each iteration $t$, we denote the collected samples as $\{s_t(s,a)\}_{(s,a) \in \mathcal{K}}$ and define a matrix $\widehat{\boldsymbol{P}}_\mathcal{K}^{(t)} \in \{0,1\}^{K \times |\mathcal{S}|}$ to be

$$\widehat{\boldsymbol{P}}_\mathcal{K}^{(t)}\left((s,a),s'\right) := \begin{cases} 1, & \text{if } s' = s_t(s,a) \\ 0, & \text{otherwise} \end{cases} \tag{15}$$

for any $(s,a) \in \mathcal{K}$ and $s' \in \mathcal{S}$. Further, we define $\widehat{\boldsymbol{P}}_t = \boldsymbol{\Lambda}\widehat{\boldsymbol{P}}_\mathcal{K}^{(t)}$. Then it is obvious to see that $\widehat{\boldsymbol{P}}_t$ has nonnegative entries and unit $\ell_1$ norm for each row due to Assumption 1, i.e. $\|\widehat{\boldsymbol{P}}_t\|_1 = 1$.

# B    Analysis of model-based RL (Proof of Theorem 1)

In this section, we will provide complete proof for Theorem 1. As a matter of fact, our proof strategy here justifies a more general version of Theorem 1 that accounts for model misspecification, as stated below.

**Theorem 3.** *Suppose that $\delta > 0$ and $\varepsilon \in (0, (1-\gamma)^{-1/2}]$. Assume that there exists a probability transition model $\widetilde{\boldsymbol{P}}$ obeying Definition 1 and Assumption 1 with feature vectors $\{\phi(s,a)\}_{(s,a) \in \mathcal{S} \times \mathcal{A}} \subset \mathbb{R}^K$ and anchor state-action pairs $\mathcal{K}$ such that*

$$\|\widetilde{\boldsymbol{P}} - \boldsymbol{P}\|_1 \leq \xi$$

*for some $\xi \geq 0$. Let $\widehat{\pi}$ be the policy returned by Algorithm 1. Assume that*

$$N \geq \frac{C \log \left(K / \left((1-\gamma)\delta\right)\right)}{(1-\gamma)^3 \varepsilon^2} \tag{16}$$

*for some sufficiently large constant $C > 0$. Then with probability exceeding $1 - \delta$,*

$$Q^\star(s,a) - Q^{\widehat{\pi}}(s,a) \leq \varepsilon + \frac{4\varepsilon_{\mathsf{opt}}}{1-\gamma} + \frac{22\xi}{(1-\gamma)^2}, \tag{17}$$

*for every state-action pair $(s,a) \in \mathcal{S} \times \mathcal{A}$.*

Theorem 3 subsumes Theorem 1 as a special case with $\xi = 0$. The remainder of this section is devoted to proving Theorem 3.

## B.1    Proof of Theorem 3

The error $\boldsymbol{Q}^{\widehat{\pi}} - \boldsymbol{Q}^\star$ can be decomposed as

$$\begin{aligned} \boldsymbol{Q}^{\widehat{\pi}} - \boldsymbol{Q}^\star &= \boldsymbol{Q}^{\widehat{\pi}} - \widehat{\boldsymbol{Q}}^{\widehat{\pi}} + \widehat{\boldsymbol{Q}}^{\widehat{\pi}} - \widehat{\boldsymbol{Q}}^\star + \widehat{\boldsymbol{Q}}^\star - \boldsymbol{Q}^\star \\ &\geq \boldsymbol{Q}^{\widehat{\pi}} - \widehat{\boldsymbol{Q}}^{\widehat{\pi}} + \widehat{\boldsymbol{Q}}^{\widehat{\pi}} - \widehat{\boldsymbol{Q}}^\star + \widehat{\boldsymbol{Q}}^{\pi^\star} - \boldsymbol{Q}^\star \\ &\geq -\left(\left\|\boldsymbol{Q}^{\widehat{\pi}} - \widehat{\boldsymbol{Q}}^{\widehat{\pi}}\right\|_\infty + \left\|\widehat{\boldsymbol{Q}}^{\widehat{\pi}} - \widehat{\boldsymbol{Q}}^\star\right\|_\infty + \left\|\widehat{\boldsymbol{Q}}^{\pi^\star} - \boldsymbol{Q}^\star\right\|_\infty\right)\mathbf{1}. \end{aligned} \tag{18}$$

For policy $\widehat{\pi}$ satisfying the condition in Theorem 1, we have $\|\widehat{\boldsymbol{Q}}^{\widehat{\pi}} - \widehat{\boldsymbol{Q}}^\star\|_\infty \leq \varepsilon_{\mathsf{opt}}$. It boils down to control $\|\boldsymbol{Q}^{\widehat{\pi}} - \widehat{\boldsymbol{Q}}^{\widehat{\pi}}\|_\infty$ and $\|\widehat{\boldsymbol{Q}}^{\pi^\star} - \boldsymbol{Q}^\star\|_\infty$.

To begin with, we can use (11) to further decompose $\left\|\boldsymbol{Q}^{\widehat{\pi}} - \widehat{\boldsymbol{Q}}^{\widehat{\pi}}\right\|_\infty$ as

$$
\begin{aligned}
\left\|\boldsymbol{Q}^{\widehat{\pi}} - \widehat{\boldsymbol{Q}}^{\widehat{\pi}}\right\|_\infty &= \left\|\left(\boldsymbol{I} - \gamma \boldsymbol{P}^{\widehat{\pi}}\right)^{-1} \boldsymbol{r} - \left(\boldsymbol{I} - \gamma \widehat{\boldsymbol{P}}^{\widehat{\pi}}\right)^{-1} \boldsymbol{r}\right\|_\infty \\
&= \left\|\left(\boldsymbol{I} - \gamma \boldsymbol{P}^{\widehat{\pi}}\right)^{-1} \left[\left(\boldsymbol{I} - \gamma \widehat{\boldsymbol{P}}^{\widehat{\pi}}\right) - \left(\boldsymbol{I} - \gamma \boldsymbol{P}^{\widehat{\pi}}\right)\right] \widehat{\boldsymbol{Q}}^{\widehat{\pi}}\right\|_\infty \\
&= \left\|\gamma \left(\boldsymbol{I} - \gamma \boldsymbol{P}^{\widehat{\pi}}\right)^{-1} \left(\boldsymbol{P} - \widehat{\boldsymbol{P}}\right) \widehat{\boldsymbol{V}}^{\widehat{\pi}}\right\|_\infty \\
&\leq \left\|\gamma \left(\boldsymbol{I} - \gamma \boldsymbol{P}^{\widehat{\pi}}\right)^{-1} \left(\boldsymbol{P} - \widehat{\boldsymbol{P}}\right) \widehat{\boldsymbol{V}}^\star\right\|_\infty + \left\|\gamma \left(\boldsymbol{I} - \gamma \boldsymbol{P}^{\widehat{\pi}}\right)^{-1} \left(\boldsymbol{P} - \widehat{\boldsymbol{P}}\right) \left(\widehat{\boldsymbol{V}}^{\widehat{\pi}} - \widehat{\boldsymbol{V}}^\star\right)\right\|_\infty \\
&\leq \left\|\gamma \left(\boldsymbol{I} - \gamma \boldsymbol{P}^{\widehat{\pi}}\right)^{-1} \left|\left(\boldsymbol{P} - \widehat{\boldsymbol{P}}\right) \widehat{\boldsymbol{V}}^\star\right|\right\|_\infty + \frac{2\gamma \varepsilon_{\text{opt}}}{1 - \gamma}. \qquad (19)
\end{aligned}
$$

Here the last inequality is due to

$$
\begin{aligned}
&\left\|\gamma \left(\boldsymbol{I} - \gamma \boldsymbol{P}^{\widehat{\pi}}\right)^{-1} \left(\boldsymbol{P} - \widehat{\boldsymbol{P}}\right) \left(\widehat{\boldsymbol{V}}^{\widehat{\pi}} - \widehat{\boldsymbol{V}}^\star\right)\right\|_\infty \\
&\qquad \leq \gamma \left\|\left(\boldsymbol{I} - \gamma \boldsymbol{P}^{\widehat{\pi}}\right)^{-1}\right\|_1 \left\|\left(\boldsymbol{P} - \widehat{\boldsymbol{P}}\right) \left(\widehat{\boldsymbol{V}}^{\widehat{\pi}} - \widehat{\boldsymbol{V}}^\star\right)\right\|_\infty \\
&\qquad \leq \gamma \left\|\left(\boldsymbol{I} - \gamma \boldsymbol{P}^{\widehat{\pi}}\right)^{-1}\right\|_1 \left(\|\boldsymbol{P}\|_1 + \left\|\widehat{\boldsymbol{P}}\right\|_1\right) \left\|\widehat{\boldsymbol{V}}^{\widehat{\pi}} - \widehat{\boldsymbol{V}}^\star\right\|_\infty \\
&\qquad \leq \frac{2\gamma \varepsilon_{\text{opt}}}{1 - \gamma},
\end{aligned}
$$

where we use the fact that $\|(\boldsymbol{I} - \gamma \widehat{\boldsymbol{P}}^{\widehat{\pi}})^{-1}\|_1 \leq 1/(1 - \gamma)$ and $\|\boldsymbol{P}\|_1 = \|\widehat{\boldsymbol{P}}\|_1 = 1$.

Similarly, for the term $\|\widehat{\boldsymbol{Q}}^{\pi^\star} - \boldsymbol{Q}^\star\|_\infty$ in (18), we have

$$
\begin{aligned}
\left\|\widehat{\boldsymbol{Q}}^{\pi^\star} - \boldsymbol{Q}^\star\right\|_\infty &= \left\|\gamma \left(\boldsymbol{I} - \gamma \boldsymbol{P}^{\pi^\star}\right)^{-1} \left(\boldsymbol{P} - \widehat{\boldsymbol{P}}\right) \widehat{\boldsymbol{V}}^{\pi^\star}\right\|_\infty \\
&\leq \left\|\gamma \left(\boldsymbol{I} - \gamma \boldsymbol{P}^{\pi^\star}\right)^{-1} \left|\left(\boldsymbol{P} - \widehat{\boldsymbol{P}}\right) \widehat{\boldsymbol{V}}^{\pi^\star}\right|\right\|_\infty. \qquad (20)
\end{aligned}
$$

As can be seen from (19) and (20), it boils down to bound $|(\boldsymbol{P} - \widehat{\boldsymbol{P}})\widehat{\boldsymbol{V}}^\star|$ and $|(\boldsymbol{P} - \widehat{\boldsymbol{P}})\widehat{\boldsymbol{V}}^{\pi^\star}|$. We have the following lemma.

**Lemma 1.** *With probability exceeding $1 - \delta$, one has*

$$
\begin{aligned}
\left|\left(\boldsymbol{P} - \widehat{\boldsymbol{P}}\right)_{s,a} \widehat{\boldsymbol{V}}^\star\right| &\leq \frac{10\xi}{1 - \gamma} + 4\sqrt{\frac{2 \log\left(4K/\delta\right)}{N}} + \frac{4 \log\left(8K/\left(\left(1 - \gamma\right)\delta\right)\right)}{\left(1 - \gamma\right)N} \\
&\quad + \sqrt{\frac{4 \log\left(8K/\left(\left(1 - \gamma\right)\delta\right)\right)}{N}} \sqrt{\mathsf{Var}_{\boldsymbol{P}_{s,a}}\left(\widehat{\boldsymbol{V}}^\star\right)}, \qquad (21)
\end{aligned}
$$

$$
\begin{aligned}
\left|\left(\boldsymbol{P} - \widehat{\boldsymbol{P}}\right)_{s,a} \widehat{\boldsymbol{V}}^{\pi^\star}\right| &\leq \frac{10\xi}{1 - \gamma} + 4\sqrt{\frac{2 \log\left(4K/\delta\right)}{N}} + \frac{4 \log\left(8K/\left(\left(1 - \gamma\right)\delta\right)\right)}{\left(1 - \gamma\right)N} \\
&\quad + \sqrt{\frac{4 \log\left(8K/\left(\left(1 - \gamma\right)\delta\right)\right)}{N}} \sqrt{\mathsf{Var}_{\boldsymbol{P}_{s,a}}\left(\widehat{\boldsymbol{V}}^{\pi^\star}\right)}. \qquad (22)
\end{aligned}
$$

*Proof.* See Appendix B.2. $\qquad\qquad\square$

Applying (21) to (19) reveals that

$$\left\| \boldsymbol{Q}^{\widehat{\pi}} - \widehat{\boldsymbol{Q}}^{\widehat{\pi}} \right\|_{\infty} \le \sqrt{\frac{4 \log\left(8K/\left((1-\gamma)\,\delta\right)\right)}{N}} \left\| \gamma \left(\boldsymbol{I} - \gamma \boldsymbol{P}^{\widehat{\pi}}\right)^{-1} \sqrt{\mathsf{Var}_{\boldsymbol{P}_{s,a}}\left(\widehat{\boldsymbol{V}}^{\star}\right)} \right\|_{\infty}$$

$$+ \frac{\gamma}{1-\gamma} \left[ 4\sqrt{\frac{2 \log\left(4K/\delta\right)}{N}} + \frac{4 \log\left(8K/\left((1-\gamma)\,\delta\right)\right)}{(1-\gamma)\,N} \right]$$

$$+ \frac{10\gamma\xi}{(1-\gamma)^2} + \frac{2\gamma\varepsilon_{\mathrm{opt}}}{1-\gamma}. \tag{23}$$

For the first term, one has

$$\sqrt{\mathsf{Var}_{\boldsymbol{P}_{s,a}}\left(\widehat{\boldsymbol{V}}^{\star}\right)} \le \sqrt{\mathsf{Var}_{\boldsymbol{P}_{s,a}}\left(\boldsymbol{V}^{\widehat{\pi}}\right)} + \sqrt{\mathsf{Var}_{\boldsymbol{P}_{s,a}}\left(\boldsymbol{V}^{\widehat{\pi}} - \widehat{\boldsymbol{V}}^{\widehat{\pi}}\right)} + \sqrt{\mathsf{Var}_{\boldsymbol{P}_{s,a}}\left(\widehat{\boldsymbol{V}}^{\widehat{\pi}} - \widehat{\boldsymbol{V}}^{\star}\right)}$$

$$\le \sqrt{\mathsf{Var}_{\boldsymbol{P}_{s,a}}\left(\boldsymbol{V}^{\widehat{\pi}}\right)} + \left\| \boldsymbol{V}^{\widehat{\pi}} - \widehat{\boldsymbol{V}}^{\widehat{\pi}} \right\|_{\infty} + \varepsilon_{\mathrm{opt}}$$

$$\le \sqrt{\mathsf{Var}_{\boldsymbol{P}_{s,a}}\left(\boldsymbol{V}^{\widehat{\pi}}\right)} + \left\| \boldsymbol{Q}^{\widehat{\pi}} - \widehat{\boldsymbol{Q}}^{\widehat{\pi}} \right\|_{\infty} + \varepsilon_{\mathrm{opt}},$$

where the first inequality comes from the fact that $\sqrt{\mathsf{Var}(X+Y)} \le \sqrt{\mathsf{Var}(X)} + \sqrt{\mathsf{Var}(Y)}$ for any random variables $X$ and $Y$. It follows that

$$\left\| \gamma \left(\boldsymbol{I} - \gamma \boldsymbol{P}^{\widehat{\pi}}\right)^{-1} \sqrt{\mathsf{Var}_{\boldsymbol{P}_{s,a}}\left(\widehat{\boldsymbol{V}}^{\star}\right)} \right\|_{\infty}$$

$$\le \left\| \gamma \left(\boldsymbol{I} - \gamma \boldsymbol{P}^{\widehat{\pi}}\right)^{-1} \sqrt{\mathsf{Var}_{\boldsymbol{P}_{s,a}}\left(\boldsymbol{V}^{\widehat{\pi}}\right)} \right\|_{\infty} + \frac{\gamma}{1-\gamma} \left( \left\| \boldsymbol{Q}^{\widehat{\pi}} - \widehat{\boldsymbol{Q}}^{\widehat{\pi}} \right\|_{\infty} + \varepsilon_{\mathrm{opt}} \right)$$

$$\le \gamma \sqrt{\frac{2}{(1-\gamma)^3}} + \frac{\gamma}{1-\gamma} \left( \left\| \boldsymbol{Q}^{\widehat{\pi}} - \widehat{\boldsymbol{Q}}^{\widehat{\pi}} \right\|_{\infty} + \varepsilon_{\mathrm{opt}} \right), \tag{24}$$

where the second inequality utilizes [3, Lemma 7].

Plugging (24) into (23) yields

$$\left\| \boldsymbol{Q}^{\widehat{\pi}} - \widehat{\boldsymbol{Q}}^{\widehat{\pi}} \right\|_{\infty} \le \sqrt{\frac{4 \log\left(8K/\left((1-\gamma)\,\delta\right)\right)}{N}} \left[ \gamma\sqrt{\frac{2}{(1-\gamma)^3}} + \frac{\gamma}{1-\gamma} \left( \left\| \boldsymbol{Q}^{\widehat{\pi}} - \widehat{\boldsymbol{Q}}^{\widehat{\pi}} \right\|_{\infty} + \varepsilon_{\mathrm{opt}} \right) \right]$$

$$+ \frac{\gamma}{1-\gamma} \left[ 4\sqrt{\frac{2 \log\left(4K/\delta\right)}{N}} + \frac{4 \log\left(8K/\left((1-\gamma)\,\delta\right)\right)}{(1-\gamma)\,N} \right] + \frac{10\gamma\xi}{(1-\gamma)^2} + \frac{2\gamma\varepsilon_{\mathrm{opt}}}{1-\gamma}.$$

Then we can rearrange terms to obtain

$$\left\| \boldsymbol{Q}^{\widehat{\pi}} - \widehat{\boldsymbol{Q}}^{\widehat{\pi}} \right\|_{\infty} \le 10\gamma \sqrt{\frac{\log\left(8K/\left((1-\gamma)\,\delta\right)\right)}{N\left(1-\gamma\right)^3}} + \frac{11\gamma\xi}{(1-\gamma)^2} + \frac{3\gamma\varepsilon_{\mathrm{opt}}}{1-\gamma} \tag{25}$$

as long as $N \ge \widetilde{C} \log(8K/((1-\gamma)\delta))/(1-\gamma)^2$ for some sufficiently large constant $\widetilde{C} > 0$.

In a similar vein, we can use (20) and (22) to obtain that

$$\left\| \widehat{\boldsymbol{Q}}^{\pi^{\star}} - \boldsymbol{Q}^{\star} \right\|_{\infty} \le 10\gamma \sqrt{\frac{\log\left(8K/\left((1-\gamma)\,\delta\right)\right)}{N\left(1-\gamma\right)^3}} + \frac{11\gamma\xi}{(1-\gamma)^2}. \tag{26}$$

Finally, we can substitute (25) and (26) into (18) to achieve

$$\boldsymbol{Q}^{\widehat{\pi}} - \boldsymbol{Q}^{\star} \ge - \left( 20\gamma \sqrt{\frac{\log\left(8K/\left((1-\gamma)\,\delta\right)\right)}{N\left(1-\gamma\right)^3}} + \frac{22\gamma\xi}{(1-\gamma)^2} + \frac{3\gamma\varepsilon_{\mathrm{opt}}}{1-\gamma} + \varepsilon_{\mathrm{opt}} \right) \mathbf{1}.$$

This result implies that

$$\boldsymbol{Q}^{\widehat{\pi}} \ge \boldsymbol{Q}^{\star} - \left( \varepsilon + \frac{22\xi}{(1-\gamma)^2} + \frac{4\varepsilon_{\mathrm{opt}}}{1-\gamma} \right) \mathbf{1},$$

as long as

$$N \ge \frac{C \log\left(8K/\left((1-\gamma)\,\delta\right)\right)}{(1-\gamma)^3 \varepsilon^2},$$

for some sufficiently large constant $C > 0$.

## B.2 Proof of Lemma 1

To prove this theorem, we invoke the idea of $s$-absorbing MDP proposed by [1]. For a state $s \in \mathcal{S}$ and a scalar $u$, we define a new MDP $M_{s,u}$ to be identical to $M$ on all the other states except $s$; on state $s$, $M_{s,u}$ is absorbing such that $P_{M_{s,u}}(s|s,a) = 1$ and $r_{M_{s,u}}(s,a) = (1-\gamma)u$ for all $a \in \mathcal{A}$. More formally, we define $P_{M_{u,s}}$ and $r_{M_{u,s}}$ as

$$P_{M_{s,u}}(s|s,a) = 1, \quad r_{M_{s,u}}(s,a) = (1-\gamma)u, \qquad \text{for all } a \in \mathcal{A},$$

$$P_{M_{s,u}}(\cdot|s',a') = P(\cdot|s',a'), \quad r_{M_{s,u}}(s,a) = r(s,a), \qquad \text{for all } s' \neq s \text{ and } a' \in \mathcal{A}.$$

To streamline notations, we will use $\boldsymbol{V}_{s,u}^{\pi} \in \mathbb{R}^{|\mathcal{S}|}$ and $\boldsymbol{V}_{s,u}^{\star} \in \mathbb{R}^{|\mathcal{S}|}$ to denote the value function of $M_{s,u}$ under policy $\pi$ and the optimal value function of $M_{s,u}$ respectively. Furthermore, we denote by $\widehat{M}_{s,u}$ the MDP whose probability transition kernel is identical to $\widehat{P}$ at all states except that state $s$ is absorbing. Similar as before, we use $\widehat{\boldsymbol{V}}_{s,u}^{\star} \in \mathbb{R}^{|\mathcal{S}|}$ to denote the optimal value function under $\widehat{M}_{s,u}$. The construction of this collection of auxiliary MDPs will facilitate our analysis by decoupling the statistical dependency between $\widehat{\boldsymbol{P}}$ and $\widehat{\pi}^{\star}$.

To begin with, we can decompose the quantity of interest as

$$
\begin{aligned}
\left| \left(\boldsymbol{P} - \widehat{\boldsymbol{P}}\right)_{s,a} \widehat{\boldsymbol{V}}^{\star} \right| &= \left| \left(\boldsymbol{P} - \widehat{\boldsymbol{P}}\right)_{s,a} \left(\widehat{\boldsymbol{V}}^{\star} - \widehat{\boldsymbol{V}}_{s,u}^{\star} + \widehat{\boldsymbol{V}}_{s,u}^{\star}\right) \right| \\
&\leq \left| \left(\boldsymbol{P} - \widehat{\boldsymbol{P}}\right)_{s,a} \widehat{\boldsymbol{V}}_{s,u}^{\star} \right| + \left| \left(\boldsymbol{P} - \widehat{\boldsymbol{P}}\right)_{s,a} \left(\widehat{\boldsymbol{V}}^{\star} - \widehat{\boldsymbol{V}}_{s,u}^{\star}\right) \right| \\
&\overset{(\mathrm{i})}{\leq} \left| \left(\boldsymbol{P} - \widetilde{\boldsymbol{P}}\right)_{s,a} \widehat{\boldsymbol{V}}_{s,u}^{\star} \right| + \left| \boldsymbol{\lambda}(s,a) \left(\widetilde{\boldsymbol{P}}_{\mathcal{K}} - \boldsymbol{P}_{\mathcal{K}}\right) \widehat{\boldsymbol{V}}_{s,u}^{\star} \right| \\
&\quad + \left| \boldsymbol{\lambda}(s,a) \left(\boldsymbol{P}_{\mathcal{K}} - \widehat{\boldsymbol{P}}_{\mathcal{K}}\right) \widehat{\boldsymbol{V}}_{s,u}^{\star} \right| + \left( \|\boldsymbol{P}_{s,a}\|_1 + \left\|\widehat{\boldsymbol{P}}_{s,a}\right\|_1 \right) \left\| \widehat{\boldsymbol{V}}^{\star} - \widehat{\boldsymbol{V}}_{s,u}^{\star} \right\|_{\infty} \\
&\leq \left\| \left(\boldsymbol{P} - \widetilde{\boldsymbol{P}}\right)_{s,a} \right\|_1 \left\| \widehat{\boldsymbol{V}}_{s,u}^{\star} \right\|_{\infty} + \|\boldsymbol{\lambda}(s,a)\|_1 \cdot \left\| \left(\widetilde{\boldsymbol{P}}_{\mathcal{K}} - \boldsymbol{P}_{\mathcal{K}}\right) \widehat{\boldsymbol{V}}_{s,u}^{\star} \right\|_{\infty} \\
&\quad + \|\boldsymbol{\lambda}(s,a)\|_1 \cdot \left\| \left(\boldsymbol{P}_{\mathcal{K}} - \widehat{\boldsymbol{P}}_{\mathcal{K}}\right) \widehat{\boldsymbol{V}}_{s,u}^{\star} \right\|_{\infty} + 2 \left\| \widehat{\boldsymbol{V}}^{\star} - \widehat{\boldsymbol{V}}_{s,u}^{\star} \right\|_{\infty} \\
&\overset{(\mathrm{ii})}{\leq} \frac{2\xi}{1-\gamma} + \max_{(s,a) \in \mathcal{K}} \left| \left(\boldsymbol{P} - \widehat{\boldsymbol{P}}\right)_{s,a} \widehat{\boldsymbol{V}}_{s,u}^{\star} \right| + 2 \left\| \widehat{\boldsymbol{V}}^{\star} - \widehat{\boldsymbol{V}}_{s,u}^{\star} \right\|_{\infty},
\end{aligned}
\tag{27}
$$

where (i) makes use of $\widetilde{\boldsymbol{P}}_{s,a} = \boldsymbol{\lambda}(s,a)\widetilde{\boldsymbol{P}}_{\mathcal{K}}$ and $\widehat{\boldsymbol{P}}_{s,a} = \boldsymbol{\lambda}(s,a)\widehat{\boldsymbol{P}}_{\mathcal{K}}$; (ii) depends on $\|\boldsymbol{P} - \widetilde{\boldsymbol{P}}\|_1 \leq \xi$, $\|\boldsymbol{\lambda}(s,a)\|_1 = 1$ and $\|\widehat{\boldsymbol{V}}_{s,u}^{\star}\|_{\infty} \leq (1-\gamma)^{-1}$. For each state $s$, the value of $u$ will be selected from a set $\mathcal{U}_s$. The choice of $\mathcal{U}_s$ will be specified later. Then for some fixed $u$ in $\mathcal{U}_s$ and fixed state-action pair $(s,a) \in \mathcal{K}$, due to the independence between $\widehat{\boldsymbol{P}}_{s,a}$ and $\widehat{\boldsymbol{V}}_{s,u}^{\star}$, we can apply Bernstein's inequality (cf. [5, Theorem 2.8.4]) conditional on $\widehat{\boldsymbol{V}}_{s,u}^{\star}$ to reveal that with probability greater than $1 - \delta/2$,

$$\left| \left(\boldsymbol{P} - \widehat{\boldsymbol{P}}\right)_{s,a} \widehat{\boldsymbol{V}}_{s,u}^{\star} \right| \leq \sqrt{\frac{2\log(4/\delta)}{N} \mathsf{Var}_{\boldsymbol{P}_{s,a}}\left(\widehat{\boldsymbol{V}}_{s,u}^{\star}\right)} + \frac{2\log(4/\delta)}{3(1-\gamma)N}. \tag{28}$$

Invoking the union bound over all the $K$ state-action pairs of $\mathcal{K}$ and all the possible values of $u$ in $\mathcal{U}_s$ demonstrate that with probability greater than $1 - \delta/2$,

$$\left| \left(\boldsymbol{P} - \widehat{\boldsymbol{P}}\right)_{s,a} \widehat{\boldsymbol{V}}_{s,u}^{\star} \right| \leq \sqrt{\frac{2\log(4K|\mathcal{U}_s|/\delta)}{N} \mathsf{Var}_{\boldsymbol{P}_{s,a}}\left(\widehat{\boldsymbol{V}}_{s,u}^{\star}\right)} + \frac{2\log(4K|\mathcal{U}_s|/\delta)}{3(1-\gamma)N}, \tag{29}$$

holds for all state-action pair $(s,a) \in \mathcal{K}$ and all $u \in \mathcal{U}_s$. Here, $\mathsf{Var}_{\boldsymbol{P}_{s,a}}(\cdot)$ is defined in (13). Then we observe that

$$
\begin{aligned}
\sqrt{\mathsf{Var}_{\boldsymbol{P}_{s,a}}\left(\widehat{\boldsymbol{V}}_{s,u}^{\star}\right)} &\leq \sqrt{\mathsf{Var}_{\boldsymbol{P}_{s,a}}\left(\widehat{\boldsymbol{V}}^{\star} - \widehat{\boldsymbol{V}}_{s,u}^{\star}\right)} + \sqrt{\mathsf{Var}_{\boldsymbol{P}_{s,a}}\left(\widehat{\boldsymbol{V}}^{\star}\right)} \\
&\leq \left\| \widehat{\boldsymbol{V}}^{\star} - \widehat{\boldsymbol{V}}_{s,u}^{\star} \right\|_{\infty} + \sqrt{\mathsf{Var}_{\boldsymbol{P}_{s,a}}\left(\widehat{\boldsymbol{V}}^{\star}\right)} \\
&\leq \left| \widehat{\boldsymbol{V}}^{\star}(s) - u \right| + \sqrt{\mathsf{Var}_{\boldsymbol{P}_{s,a}}\left(\widehat{\boldsymbol{V}}^{\star}\right)},
\end{aligned}
\tag{30}
$$

where (i) is due to $\sqrt{\operatorname{Var}_{\boldsymbol{P}_{s,a}}(\boldsymbol{V}_1 + \boldsymbol{V}_2)} \leq \sqrt{\operatorname{Var}_{\boldsymbol{P}_{s,a}}(\boldsymbol{V}_1)} + \sqrt{\operatorname{Var}_{\boldsymbol{P}_{s,a}}(\boldsymbol{V}_2)}$ and (ii) holds since

$$\left\| \widehat{\boldsymbol{V}}^\star - \widehat{\boldsymbol{V}}^\star_{s,u} \right\|_\infty = \left\| \widehat{\boldsymbol{V}}^\star_{s,\widehat{\boldsymbol{V}}^\star(s)} - \widehat{\boldsymbol{V}}^\star_{s,u} \right\|_\infty \leq \left| \widehat{\boldsymbol{V}}^\star(s) - u \right|, \tag{31}$$

whose proof can be found in [1, Lemma 8 and 9].

By substituting (29), (30) and (31) into (27), we arrive at

$$\left| \left( \boldsymbol{P} - \widehat{\boldsymbol{P}} \right)_{s,a} \widehat{\boldsymbol{V}}^\star \right| \leq \frac{2\xi}{1-\gamma} + \left| \widehat{\boldsymbol{V}}^\star(s) - u \right| \left( 2 + \sqrt{\frac{2\log\left(4K\left|\mathcal{U}_s\right|/\delta\right)}{N}} \right)$$
$$+ \sqrt{\frac{2\log\left(4K\left|\mathcal{U}_s\right|/\delta\right)}{N}} \sqrt{\operatorname{Var}_{\boldsymbol{P}_{s,a}}\left(\widehat{\boldsymbol{V}}^\star\right)} + \frac{2\log\left(4K\left|\mathcal{U}_s\right|/\delta\right)}{3\left(1-\gamma\right)N}. \tag{32}$$

Then it boils down to determining $\mathcal{U}_s$. The coarse bounds of $\widehat{\boldsymbol{Q}}^{\pi^\star}$ and $\widehat{\boldsymbol{Q}}^\star$ in the following lemma provide a guidance on the choice of $\mathcal{U}_s$.

**Lemma 2.** *For $\delta \in (0,1)$, with probability exceeding $1 - \delta/2$ one has*

$$\left\| \boldsymbol{Q}^\star - \widehat{\boldsymbol{Q}}^{\pi^\star} \right\|_\infty \leq \frac{\gamma}{1-\gamma} \sqrt{\frac{\log\left(4K/\delta\right)}{2N\left(1-\gamma\right)^2}} + \frac{2\gamma\xi}{\left(1-\gamma\right)^2}, \tag{33}$$

$$\left\| \boldsymbol{Q}^\star - \widehat{\boldsymbol{Q}}^\star \right\|_\infty \leq \frac{\gamma}{1-\gamma} \sqrt{\frac{\log\left(4K/\delta\right)}{2N\left(1-\gamma\right)^2}} + \frac{2\gamma\xi}{\left(1-\gamma\right)^2}. \tag{34}$$

*Proof.* See Appendix B.3. $\qquad\square$

This inspires us to choose $\mathcal{U}_s$ to be the set consisting of equidistant points in $[\boldsymbol{V}^\star(s) - R(\delta), \boldsymbol{V}^\star(s) + R(\delta)]$ with $|U_s| = \lceil 1/(1-\gamma)^2 \rceil$ and

$$R(\delta) := \frac{\gamma}{1-\gamma} \sqrt{\frac{\log\left(4K/\delta\right)}{2N\left(1-\gamma\right)^2}} + \frac{2\gamma\xi}{\left(1-\gamma\right)^2}.$$

Since $\|\boldsymbol{V}^\star - \widehat{\boldsymbol{V}}^\star\|_\infty \leq \|\boldsymbol{Q}^\star - \widehat{\boldsymbol{Q}}^\star\|_\infty$, Lemma 2 implies that $\widehat{\boldsymbol{V}}^\star(s) \in [\boldsymbol{V}^\star(s) - R(\delta), \boldsymbol{V}^\star(s) + R(\delta)]$ with probability over $1 - \delta/2$. Hence, we have

$$\min_{u \in \mathcal{U}_s} \left| \widehat{\boldsymbol{V}}^\star(s) - u \right| \leq \frac{2R(\delta)}{|U_s| + 1} \leq 2\gamma \sqrt{\frac{2\log\left(4K/\delta\right)}{N}} + 4\gamma\xi. \tag{35}$$

Consequently, with probability exceeding $1 - \delta$, one has

$$\left| \left( \boldsymbol{P} - \widehat{\boldsymbol{P}} \right)_{s,a} \widehat{\boldsymbol{V}}^\star \right| \overset{(i)}{\leq} \frac{2\xi}{1-\gamma} + \min_{u \in \mathcal{U}_s} \left| \widehat{\boldsymbol{V}}^\star(s) - u \right| \left( 2 + \sqrt{\frac{2\log\left(4K\left|\mathcal{U}_s\right|/\delta\right)}{N}} \right)$$
$$+ \sqrt{\frac{2\log\left(4K\left|\mathcal{U}_s\right|/\delta\right)}{N}} \sqrt{\operatorname{Var}_{\boldsymbol{P}_{s,a}}\left(\widehat{\boldsymbol{V}}^\star\right)} + \frac{2\log\left(4K\left|\mathcal{U}_s\right|/\delta\right)}{3\left(1-\gamma\right)N}$$
$$\overset{(ii)}{\leq} \frac{2\xi}{1-\gamma} + \left( 2\gamma\sqrt{\frac{2\log\left(4K/\delta\right)}{N}} + 4\gamma\xi \right) \left( 2 + \sqrt{\frac{4\log\left(8K/\left(\left(1-\gamma\right)\delta\right)\right)}{N}} \right)$$
$$+ \sqrt{\frac{4\log\left(8K/\left(\left(1-\gamma\right)\delta\right)\right)}{N}} \sqrt{\operatorname{Var}_{\boldsymbol{P}_{s,a}}\left(\widehat{\boldsymbol{V}}^\star\right)} + \frac{2\log\left(8K/\left(\left(1-\gamma\right)\delta\right)\right)}{3\left(1-\gamma\right)N}$$
$$\leq \frac{10\xi}{1-\gamma} + 4\sqrt{\frac{2\log\left(4K/\delta\right)}{N}} + \frac{4\log\left(8K/\left(\left(1-\gamma\right)\delta\right)\right)}{\left(1-\gamma\right)N}$$
$$+ \sqrt{\frac{4\log\left(8K/\left(\left(1-\gamma\right)\delta\right)\right)}{N}} \sqrt{\operatorname{Var}_{\boldsymbol{P}_{s,a}}\left(\widehat{\boldsymbol{V}}^\star\right)},$$

where (i) follows from (32) and (ii) utilizes (35). This finishes the proof for the first inequality. The second inequality can be proved in a similar way and is omitted here for brevity.

## B.3 Proof of Lemma 2

To begin with, one has

$$
\begin{aligned}
\left\|\left(\widehat{\boldsymbol{P}}-\boldsymbol{P}\right)\boldsymbol{V}^\star\right\|_\infty &\leq \left\|\boldsymbol{\Lambda}\left(\widehat{\boldsymbol{P}}_\mathcal{K}-\boldsymbol{P}_\mathcal{K}\right)\boldsymbol{V}^\star\right\|_\infty + \left\|\boldsymbol{\Lambda}\left(\boldsymbol{P}_\mathcal{K}-\widetilde{\boldsymbol{P}}_\mathcal{K}\right)\boldsymbol{V}^\star\right\|_\infty + \left\|\left(\widetilde{\boldsymbol{P}}-\boldsymbol{P}\right)\boldsymbol{V}^\star\right\|_\infty \\
&\leq \|\boldsymbol{\Lambda}\|_1\left\|\left(\widehat{\boldsymbol{P}}_\mathcal{K}-\boldsymbol{P}_\mathcal{K}\right)\boldsymbol{V}^\star\right\|_\infty + \|\boldsymbol{\Lambda}\|_1\left\|\left(\boldsymbol{P}_\mathcal{K}-\widetilde{\boldsymbol{P}}_\mathcal{K}\right)\boldsymbol{V}^\star\right\|_\infty + \left\|\widetilde{\boldsymbol{P}}-\boldsymbol{P}\right\|_1\|\boldsymbol{V}^\star\|_\infty \\
&\leq \left\|\left(\widehat{\boldsymbol{P}}_\mathcal{K}-\boldsymbol{P}_\mathcal{K}\right)\boldsymbol{V}^\star\right\|_\infty + \frac{2\xi}{1-\gamma},
\end{aligned} \tag{36}
$$

where the first line uses $\widehat{\boldsymbol{P}} = \boldsymbol{\Lambda}\widehat{\boldsymbol{P}}_\mathcal{K}$ and $\widetilde{\boldsymbol{P}} = \boldsymbol{\Lambda}\widetilde{\boldsymbol{P}}_\mathcal{K}$; the last inequality comes from the facts that $\|\widetilde{\boldsymbol{P}}-\boldsymbol{P}\|_1 \leq \xi$, $\|\boldsymbol{\Lambda}\|_1 = 1$ and $\|\boldsymbol{V}^\star\|_\infty \leq (1-\gamma)^{-1}$. Then we turn to bound $\|(\widehat{\boldsymbol{P}}_\mathcal{K}-\boldsymbol{P}_\mathcal{K})\boldsymbol{V}^\star\|_\infty$. In view of (4), Hoeffding's inequality (cf. [5, Theorem 2.2.6]) implies that for $(s,a) \in \mathcal{K}$,

$$
\mathbb{P}\left(\left|\left(\widehat{\boldsymbol{P}}-\boldsymbol{P}\right)_{s,a}\boldsymbol{V}^\star\right| \geq t\right) \leq 2\exp\left(-\frac{2t^2}{\|\boldsymbol{V}^\star\|_\infty^2/N}\right).
$$

Hence by the standard union bound argument we have

$$
\left\|\left(\widehat{\boldsymbol{P}}_\mathcal{K}-\boldsymbol{P}_\mathcal{K}\right)\boldsymbol{V}^\star\right\|_\infty \leq \sqrt{\frac{\|\boldsymbol{V}^\star\|_\infty^2\log\left(4K/\delta\right)}{2N}} \leq \sqrt{\frac{\log\left(4K/\delta\right)}{2N\left(1-\gamma\right)^2}}, \tag{37}
$$

with probability over $1-\delta/2$.

1. Now we are ready to bound $\boldsymbol{Q}^{\pi^\star} - \widehat{\boldsymbol{Q}}^{\pi^\star}$. One has

$$
\begin{aligned}
\boldsymbol{Q}^{\pi^\star} - \widehat{\boldsymbol{Q}}^{\pi^\star} &= \left(\boldsymbol{I}-\gamma\boldsymbol{P}^{\pi^\star}\right)^{-1}\boldsymbol{r} - \left(\boldsymbol{I}-\gamma\widehat{\boldsymbol{P}}^{\pi^\star}\right)^{-1}\boldsymbol{r} \\
&= \left(\boldsymbol{I}-\gamma\widehat{\boldsymbol{P}}^{\pi^\star}\right)^{-1}\left(\left(\boldsymbol{I}-\gamma\widehat{\boldsymbol{P}}^{\pi^\star}\right)-\left(\boldsymbol{I}-\gamma\boldsymbol{P}^{\pi^\star}\right)\right)\boldsymbol{Q}^{\pi^\star} \\
&= \gamma\left(\boldsymbol{I}-\gamma\widehat{\boldsymbol{P}}^{\pi^\star}\right)^{-1}\left(\boldsymbol{P}^{\pi^\star}-\widehat{\boldsymbol{P}}^{\pi^\star}\right)\boldsymbol{Q}^{\pi^\star} \\
&= \gamma\left(\boldsymbol{I}-\gamma\widehat{\boldsymbol{P}}^{\pi^\star}\right)^{-1}\left(\boldsymbol{P}-\widehat{\boldsymbol{P}}\right)\boldsymbol{V}^{\pi^\star},
\end{aligned}
$$

where the first equality makes use of (11). Then we take (36) and (37) collectively to achieve

$$
\begin{aligned}
\left\|\gamma\left(\boldsymbol{I}-\gamma\widehat{\boldsymbol{P}}^{\pi^\star}\right)^{-1}\left(\boldsymbol{P}-\widehat{\boldsymbol{P}}\right)\boldsymbol{V}^\star\right\|_\infty &\leq \gamma\sum_{i=0}^{\infty}\left\|\gamma^i\left(\widehat{\boldsymbol{P}}^{\pi^\star}\right)^i\left(\boldsymbol{P}-\widehat{\boldsymbol{P}}\right)\boldsymbol{V}^\star\right\|_\infty \\
&\leq \gamma\sum_{i=0}^{\infty}\gamma^i\left\|\left(\widehat{\boldsymbol{P}}^{\pi^\star}\right)^i\right\|_1\left\|\left(\boldsymbol{P}-\widehat{\boldsymbol{P}}\right)\boldsymbol{V}^\star\right\|_\infty \\
&\leq \frac{\gamma}{1-\gamma}\sqrt{\frac{\log\left(4K/\delta\right)}{2N\left(1-\gamma\right)^2}} + \frac{2\gamma\xi}{\left(1-\gamma\right)^2},
\end{aligned}
$$

where the last line comes from the fact that for all $i \geq 1$, $(\widehat{\boldsymbol{P}}^{\pi^\star})^i$ is a probability transition matrix so that $\|(\widehat{\boldsymbol{P}}^{\pi^\star})^i\|_1 = 1$. This justifies the first inequality (33).

2. In terms of the second one, [1, Section A.4] implies that

$$
\left\|\boldsymbol{Q}^\star - \widehat{\boldsymbol{Q}}^\star\right\|_\infty \leq \frac{\gamma}{1-\gamma}\left\|\left(\boldsymbol{P}-\widehat{\boldsymbol{P}}\right)\boldsymbol{V}^\star\right\|_\infty.
$$

Substitution of (36) and (37) into the above inequality yields

$$
\left\|\boldsymbol{Q}^\star - \widehat{\boldsymbol{Q}}^\star\right\|_\infty \leq \frac{\gamma}{1-\gamma}\sqrt{\frac{\log\left(4K/\delta\right)}{2N\left(1-\gamma\right)^2}} + \frac{2\gamma\xi}{\left(1-\gamma\right)^2}.
$$

# C Analysis of Q-learning (Proof of Theorem 2)

In this section, we will provide complete proof for Theorem 2. We actually prove a more general version of Theorem 2 that takes model misspecification into consideration, as stated below.

**Theorem 4.** *Consider any $\delta \in (0,1)$ and $\varepsilon \in (0,1]$. Suppose that there exists a probability transition model $\widetilde{P}$ obeying Definition 1 and Assumption 1 with feature vectors $\{\phi(s,a)\}_{(s,a)\in\mathcal{S}\times\mathcal{A}} \subset \mathbb{R}^K$ and anchor state-action pairs $\mathcal{K}$ such that*

$$\|\widetilde{P} - P\|_1 \leq \xi$$

*for some $\xi \geq 0$. Assume that the initialization obeys $0 \leq Q_0(s,a) \leq \frac{1}{1-\gamma}$ for any $(s,a) \in \mathcal{S} \times \mathcal{A}$ and for any $0 \leq t \leq T$, the learning rates satisfy*

$$\frac{1}{1 + \frac{c_1(1-\gamma)T}{\log^2 T}} \leq \eta_t \leq \frac{1}{1 + \frac{c_2(1-\gamma)t}{\log^2 T}}, \tag{38}$$

*for some sufficiently small universal constants $c_1 \geq c_2 > 0$. Suppose that the total number of iterations $T$ exceeds*

$$T \geq \frac{C_3 \log (KT/\delta) \log^4 T}{(1-\gamma)^4 \varepsilon^2}, \tag{39}$$

*for some sufficiently large universal constant $C_3 > 0$. If there exists a linear probability transition model $\widetilde{P}$ satisfying Assumption 1 with feature vectors $\{\phi(s,a)\}_{(s,a)\in\mathcal{S}\times\mathcal{A}}$ such that $\|\widetilde{P} - P\|_1 \leq \xi$, then with probability exceeding $1 - \delta$, the output $Q_T$ of Algorithm 2 satisfies*

$$\max_{(s,a)\in\mathcal{S}\times\mathcal{A}} |Q_T(s,a) - Q^\star(s,a)| \leq \varepsilon + \frac{6\gamma\xi}{(1-\gamma)^2}, \tag{40}$$

*for some constant $C_4 > 0$. In addition, let $\pi_T$ (resp. $V_T$) to be the policy (resp. value function) induced by $Q_T$, then one has*

$$\max_{s\in\mathcal{S}} |V^{\pi_T}(s) - V^\star(s)| \leq \frac{2\gamma}{1-\gamma} \left(\varepsilon + \frac{6\gamma\xi}{(1-\gamma)^2}\right). \tag{41}$$

Theorem 4 subsumes Theorem 2 as a special case with $\xi = 0$. The remainder of this section is devoted to proving Theorem 4.

## C.1 Proof of Theorem 4

First we show that (41) can be easily obtained from (40). Since [49] gives rise to

$$\|V^{\pi_T} - V^\star\|_\infty \leq \frac{2\gamma\|V_T - V^\star\|_\infty}{1-\gamma},$$

we have

$$\|V^{\pi_T} - V^\star\|_\infty \leq \frac{2\gamma\|Q_T - Q^\star\|_\infty}{1-\gamma},$$

due to $\|V_T - V^\star\|_\infty \leq \|Q_T - Q^\star\|_\infty$. Then (41) follows directly from (40).

Therefore, we are left to justify (40). To start with, we consider the update rule

$$Q_t = (1 - \eta_t) Q_{t-1} + \eta_t \left(r + \gamma \widehat{P}_t V_{t-1}\right).$$

By defining the error term $\Delta_t := Q_t - Q^\star$, we can decompose $\Delta_t$ into

$$\begin{aligned}
\Delta_t &= (1 - \eta_t) Q_{t-1} + \eta_t \left(r + \gamma \widehat{P}_t V_{t-1}\right) - Q^\star \\
&= (1 - \eta_t)(Q_{t-1} - Q^\star) + \eta_t \left(r + \gamma \widehat{P}_t V_{t-1} - Q^\star\right) \\
&= (1 - \eta_t)(Q_{t-1} - Q^\star) + \gamma\eta_t \left(\widehat{P}_t V_{t-1} - P V^\star\right) \\
&= (1 - \eta_t) \Delta_{t-1} + \gamma\eta_t \Lambda \left(\widehat{P}_{\mathcal{K}}^{(t)} - P_{\mathcal{K}}\right) V_{t-1} + \gamma\eta_t \Lambda P_{\mathcal{K}}(V_{t-1} - V^\star) \\
&\quad + \gamma\eta_t (\Lambda P_{\mathcal{K}} - P) V^\star. \tag{42}
\end{aligned}$$

Here in the penultimate equality, we make use of $\boldsymbol{Q}^\star = \boldsymbol{r} + \gamma \boldsymbol{P} \boldsymbol{V}^\star$; and the last equality comes from $\widehat{\boldsymbol{P}}_t = \boldsymbol{\Lambda} \widehat{\boldsymbol{P}}_{\mathcal{K}}^{(t)}$ which is defined in (15). It is straightforward to check that $\boldsymbol{\Lambda} \boldsymbol{P}_{\mathcal{K}}$ is also a probability transition matrix. We denote by $\overline{\boldsymbol{P}} = \boldsymbol{\Lambda} \boldsymbol{P}_{\mathcal{K}}$ hereafter. The third term in the decomposition above can be upper and lower bounded by

$$\overline{\boldsymbol{P}}\left(\boldsymbol{V}_{t-1} - \boldsymbol{V}^\star\right) = \overline{\boldsymbol{P}}^{\pi^{t-1}} \boldsymbol{Q}_{t-1} - \overline{\boldsymbol{P}}^{\pi^\star} \boldsymbol{Q}^\star \le \overline{\boldsymbol{P}}^{\pi^{t-1}} \boldsymbol{Q}_{t-1} - \overline{\boldsymbol{P}}^{\pi^{t-1}} \boldsymbol{Q}^\star = \overline{\boldsymbol{P}}^{\pi^{t-1}} \boldsymbol{\Delta}_{t-1},$$

and

$$\overline{\boldsymbol{P}}\left(\boldsymbol{V}_{t-1} - \boldsymbol{V}^\star\right) = \overline{\boldsymbol{P}}^{\pi^{t-1}} \boldsymbol{Q}_{t-1} - \overline{\boldsymbol{P}}^{\pi^\star} \boldsymbol{Q}^\star \ge \overline{\boldsymbol{P}}^{\pi^\star} \boldsymbol{Q}_{t-1} - \overline{\boldsymbol{P}}^{\pi^\star} \boldsymbol{Q}^\star = \overline{\boldsymbol{P}}^{\pi^\star} \boldsymbol{\Delta}_{t-1}.$$

Plugging these bounds into (42) yields

$$\boldsymbol{\Delta}_t \le (1 - \eta_t) \boldsymbol{\Delta}_{t-1} + \gamma \eta_t \boldsymbol{\Lambda} \left(\widehat{\boldsymbol{P}}_{\mathcal{K}}^{(t)} - \boldsymbol{P}_{\mathcal{K}}\right) \boldsymbol{V}_{t-1} + \gamma \eta_t \overline{\boldsymbol{P}}^{\pi^{t-1}} \boldsymbol{\Delta}_{t-1} + \gamma \eta_t \left(\boldsymbol{\Lambda} \boldsymbol{P}_{\mathcal{K}} - \boldsymbol{P}\right) \boldsymbol{V}^\star,$$

$$\boldsymbol{\Delta}_t \ge (1 - \eta_t) \boldsymbol{\Delta}_{t-1} + \gamma \eta_t \boldsymbol{\Lambda} \left(\widehat{\boldsymbol{P}}_{\mathcal{K}}^{(t)} - \boldsymbol{P}_{\mathcal{K}}\right) \boldsymbol{V}_{t-1} + \gamma \eta_t \overline{\boldsymbol{P}}^{\pi^\star} \boldsymbol{\Delta}_{t-1} + \gamma \eta_t \left(\boldsymbol{\Lambda} \boldsymbol{P}_{\mathcal{K}} - \boldsymbol{P}\right) \boldsymbol{V}^\star.$$

Repeatedly invoking these two recursive relations leads to

$$\boldsymbol{\Delta}_t \le \eta_0^{(t)} \boldsymbol{\Delta}_0 + \sum_{i=1}^{t} \eta_i^{(t)} \gamma \left(\overline{\boldsymbol{P}}^{\pi^{t-1}} \boldsymbol{\Delta}_{t-1} + \boldsymbol{\Lambda} \left(\widehat{\boldsymbol{P}}_{\mathcal{K}}^{(t)} - \boldsymbol{P}_{\mathcal{K}}\right) \boldsymbol{V}_{t-1} + \left(\boldsymbol{\Lambda} \boldsymbol{P}_{\mathcal{K}} - \boldsymbol{P}\right) \boldsymbol{V}^\star\right), \quad (43)$$

$$\boldsymbol{\Delta}_t \ge \eta_0^{(t)} \boldsymbol{\Delta}_0 + \sum_{i=1}^{t} \eta_i^{(t)} \gamma \left(\overline{\boldsymbol{P}}^{\pi^\star} \boldsymbol{\Delta}_{t-1} + \boldsymbol{\Lambda} \left(\widehat{\boldsymbol{P}}_{\mathcal{K}}^{(t)} - \boldsymbol{P}_{\mathcal{K}}\right) \boldsymbol{V}_{t-1} + \left(\boldsymbol{\Lambda} \boldsymbol{P}_{\mathcal{K}} - \boldsymbol{P}\right) \boldsymbol{V}^\star\right), \quad (44)$$

where

$$\eta_i^{(t)} := \begin{cases} \prod_{j=1}^{t} (1 - \eta_j), & \text{if } i = 0, \\ \eta_i \prod_{j=i+1}^{t} (1 - \eta_j), & \text{if } 0 < i < t, \\ \eta_t, & \text{if } i = t. \end{cases}$$

Here we adopt the same notations as [4].

To begin with, we consider the upper bound (43). It can be further decomposed as

$$\boldsymbol{\Delta}_t \le \eta_0^{(t)} \boldsymbol{\Delta}_0 + \underbrace{\sum_{i=1}^{(1-\alpha)t} \eta_i^{(t)} \gamma \left(\overline{\boldsymbol{P}}^{\pi^{t-1}} \boldsymbol{\Delta}_{t-1} + \boldsymbol{\Lambda} \left(\widehat{\boldsymbol{P}}_{\mathcal{K}}^{(t)} - \boldsymbol{P}_{\mathcal{K}}\right) \boldsymbol{V}_{t-1}\right)}_{=:\boldsymbol{\theta}_t}$$

$$+ \underbrace{\sum_{i=(1-\alpha)t+1}^{t} \eta_i^{(t)} \gamma \boldsymbol{\Lambda} \left(\widehat{\boldsymbol{P}}_{\mathcal{K}}^{(t)} - \boldsymbol{P}_{\mathcal{K}}\right) \boldsymbol{V}_{i-1}}_{=:\boldsymbol{\nu}_t}$$

$$+ \underbrace{\sum_{i=1}^{t} \eta_i^{(t)} \gamma \left(\boldsymbol{\Lambda} \boldsymbol{P}_{\mathcal{K}} - \boldsymbol{P}\right) \boldsymbol{V}^\star}_{=:\boldsymbol{\omega}_t} + \sum_{i=(1-\alpha)t+1}^{t} \eta_i^{(t)} \gamma \overline{\boldsymbol{P}}^{\pi^{t-1}} \boldsymbol{\Delta}_{i-1}, \quad (45)$$

where we define $\alpha := C_4 (1 - \gamma) / \log T$ for some constant $C_4 > 0$. Next, we turn to bound $\boldsymbol{\theta}_t$ and $\boldsymbol{\nu}_t$ respectively for any $t$ satisfying $\frac{T}{c_2 \log \frac{1}{1-\gamma}} \le t \le T$ with stepsize choice (8).

**Bounding $\boldsymbol{\omega}_t$.** It is straightforward to bound

$$\|\boldsymbol{\omega}_t\|_\infty \overset{\text{(i)}}{=} \|\gamma \left(\boldsymbol{\Lambda} \boldsymbol{P}_{\mathcal{K}} - \boldsymbol{P}\right) \boldsymbol{V}^\star\|_\infty$$

$$\overset{\text{(ii)}}{\le} \gamma \left(\|\boldsymbol{\Lambda}\|_1 \left\|\left(\boldsymbol{P}_{\mathcal{K}} - \widetilde{\boldsymbol{P}}_{\mathcal{K}}\right) \boldsymbol{V}^\star\right\|_\infty + \left\|\left(\widetilde{\boldsymbol{P}} - \boldsymbol{P}\right) \boldsymbol{V}^\star\right\|_\infty\right)$$

$$\overset{\text{(iii)}}{\le} \frac{2\gamma\xi}{1 - \gamma},$$

where the first equality comes from the fact that $\sum_{i=1}^{t} \eta_i^{(t)} = 1$ [4, Equation (40)]; the second inequality utilizes $\widetilde{\boldsymbol{P}} = \boldsymbol{\Lambda} \widetilde{\boldsymbol{P}}_{\mathcal{K}}$; the last line uses the facts that $\|\boldsymbol{\Lambda}\|_1 = 1$, $\|\boldsymbol{V}^\star\|_\infty \le (1 - \gamma)^{-1}$ and $\|\widetilde{\boldsymbol{P}}_{\mathcal{K}} - \boldsymbol{P}_{\mathcal{K}}\|_1 \le \|\widetilde{\boldsymbol{P}} - \boldsymbol{P}\|_1 \le \xi$.

**Bounding $\boldsymbol{\theta}_t$.** By similar derivation as Step 1 in [4, Appendix A.2], we have

$$\|\boldsymbol{\theta}_t\|_\infty \leq \eta_0^{(t)} \|\boldsymbol{\Delta}_0\|_\infty + t \max_{1 \leq i \leq (1-\alpha)t} \eta_i^{(t)} \max_{1 \leq i \leq (1-\alpha)t} \left( \left\|\overline{\boldsymbol{P}}^{\pi^{t-1}} \boldsymbol{\Delta}_{i-1} \right\|_\infty + \left\| \boldsymbol{\Lambda} \widehat{\boldsymbol{P}}_{\mathcal{K}}^{(t)} \boldsymbol{V}_{i-1} \right\|_\infty + \|\boldsymbol{\Lambda} \boldsymbol{P}_{\mathcal{K}} \boldsymbol{V}_{i-1}\|_\infty \right)$$

$$\overset{(i)}{\leq} \eta_0^{(t)} \|\boldsymbol{\Delta}_0\|_\infty + t \max_{1 \leq i \leq (1-\alpha)t} \eta_i^{(t)} \max_{1 \leq i \leq (1-\alpha)t} \left( \|\boldsymbol{\Delta}_{i-1}\|_\infty + 2\|\boldsymbol{V}_{i-1}\|_\infty \right)$$

$$\overset{(ii)}{\leq} \frac{1}{2T^2} \cdot \frac{1}{1-\gamma} + \frac{1}{2T^2} \cdot t \cdot \frac{3}{1-\gamma}$$

$$\leq \frac{2}{(1-\gamma)T}, \tag{46}$$

where (i) is due to the fact that $\|\overline{\boldsymbol{P}}^{\pi^{t-1}}\|_1 = \|\boldsymbol{\Lambda}\widehat{\boldsymbol{P}}_{\mathcal{K}}^{(t)}\|_1 = \|\boldsymbol{\Lambda}\boldsymbol{P}_{\mathcal{K}}\|_1 = 1$ and (ii) comes from [4, Equation (39a)].

**Bounding $\boldsymbol{\nu}_t$.** To control the second term, we apply the following Freedman's inequality.

**Lemma 3** (Freedman's Inequality)**.** *Consider a real-valued martingale $\{Y_k : k = 0, 1, 2, \cdots\}$ with difference sequence $\{X_k : k = 1, 2, 3, \cdots\}$. Assume that the difference sequence is uniformly bounded:*

$$|X_k| \leq R \qquad and \qquad \mathbb{E}\left[X_k | \{X_j\}_{j=1}^{k-1}\right] = 0 \qquad for~all~k \geq 1.$$

*Let*

$$S_n := \sum_{k=1}^{n} X_i, \qquad T_n := \sum_{k=1}^{n} \mathsf{Var}\left\{ X_k | \{X_j\}_{j=1}^{k-1} \right\}.$$

*Then for any given $\sigma^2 \geq 0$, one has*

$$\mathbb{P}\left(|S_n| \geq \tau~and~T_n \leq \sigma^2\right) \leq 2\exp\left(-\frac{\tau^2/2}{\sigma^2 + R\tau/3}\right).$$

*In addition, suppose that $W_n \leq \sigma^2$ holds deterministically. For any positive integer $K \geq 1$, with probability at least $1 - \delta$ one has*

$$|S_n| \leq \sqrt{8\max\left\{T_n, \frac{\sigma^2}{2^K}\right\}\log\frac{2K}{\delta}} + \frac{4}{3}R\log\frac{2K}{\delta}.$$

*Proof.* See [4, Theorem 4]. $\qquad\qquad\qquad\qquad\qquad\qquad\qquad\qquad\qquad\qquad\qquad\square$

To apply this inequality, we can express $\boldsymbol{\nu}_t$ as

$$\boldsymbol{\nu}_t := \sum_{i=(1-\alpha)t+1}^{t} \boldsymbol{x}_i,$$

with

$$\boldsymbol{x}_i := \eta_i^{(t)} \gamma \boldsymbol{\Lambda} \left( \widehat{\boldsymbol{P}}_{\mathcal{K}}^{(t)} - \boldsymbol{P}_{\mathcal{K}} \right) \boldsymbol{V}_{i-1}, \quad \text{and} \quad \mathbb{E}\left[\boldsymbol{x}_i | \boldsymbol{V}_{i-1}, \cdots, \boldsymbol{V}_0\right] = \boldsymbol{0}. \tag{47}$$

1. In order to calculate bound $R$ in Lemma 3, one has

$$B := \max_{(1-\alpha)t < t \leq t} \|\boldsymbol{x}_i\|_\infty \leq \max_{(1-\alpha)t < t \leq t} \left\| \eta_i^{(t)} \boldsymbol{\Lambda} \left( \widehat{\boldsymbol{P}}_{\mathcal{K}}^{(t)} - \boldsymbol{P}_{\mathcal{K}} \right) \boldsymbol{V}_{i-1} \right\|_\infty$$

$$\leq \max_{(1-\alpha)t < t \leq t} \eta_i^{(t)} \left( \left\| \boldsymbol{\Lambda}\widehat{\boldsymbol{P}}_{\mathcal{K}}^{(t)} \right\|_1 + \|\boldsymbol{\Lambda}\boldsymbol{P}_{\mathcal{K}}\|_1 \right) \|\boldsymbol{V}_{i-1}\|_\infty$$

$$\leq \max_{(1-\alpha)t < t \leq t} \eta_i^{(t)} \cdot \frac{2}{1-\gamma} \leq \frac{4\log^4 T}{(1-\gamma)^2 T},$$

where the last inequality comes from [4, Eqn (39b)] and the fact that $\|\boldsymbol{V}_{i-1}\|_\infty \leq \frac{1}{1-\gamma}$.

2. Then regarding the variance term, we claim for the moment that

$$
\boldsymbol{W}_t := \sum_{i=(1-\alpha)t+1}^{t} \text{diag}\left(\text{Var}\left(\boldsymbol{x}_i | \boldsymbol{V}_{i-1}, \cdots, \boldsymbol{V}_0\right)\right)
$$

$$
\leq \gamma^2 \sum_{i=(1-\alpha)t+1}^{t} \left(\eta_i^{(t)}\right)^2 \text{Var}_{\overline{\boldsymbol{P}}}\left(\boldsymbol{V}_{i-1}\right). \tag{48}
$$

Then we have

$$
\boldsymbol{W}_t \leq \max_{(1-\alpha)t \leq i \leq t} \eta_i^{(t)} \left( \sum_{i=(1-\alpha)t+1}^{t} \eta_i^{(t)} \right) \max_{(1-\alpha)t \leq i < t} \text{Var}_{\overline{\boldsymbol{P}}}\left(\boldsymbol{V}_i\right)
$$

$$
\leq \frac{2 \log^4 T}{(1-\gamma)T} \max_{(1-\alpha)t \leq i < t} \text{Var}_{\overline{\boldsymbol{P}}}\left(\boldsymbol{V}_i\right), \tag{49}
$$

where the second line comes from [4, Eqns (39b), (40)]. A trivial upper bound for $\boldsymbol{W}_t$ is

$$
|\boldsymbol{W}_t| \leq \frac{2 \log^4 T}{(1-\gamma)T} \cdot \frac{1}{(1-\gamma)^2} \mathbf{1} = \frac{2 \log^4 T}{(1-\gamma)^3 T} \mathbf{1},
$$

which uses the fact that $\text{Var}_{\boldsymbol{P}}(\boldsymbol{V}_i) \leq \|\boldsymbol{V}_i\|_\infty^2 \leq 1/(1-\gamma)^2$.

Then, we invoke Lemma 3 with $K = \left\lceil 2 \log_2 \frac{1}{1-\gamma} \right\rceil$ and apply the union bound argument over $\mathcal{K}$ to arrive at

$$
|\boldsymbol{\nu}_t| \leq \sqrt{8 \left(\boldsymbol{W}_t + \frac{\sigma^2}{2^K}\mathbf{1}\right) \log \frac{8KT \log \frac{1}{1-\gamma}}{\delta}} + \frac{4}{3} B \log \frac{8KT \log \frac{1}{1-\gamma}}{\delta} \mathbf{1}
$$

$$
\leq \sqrt{8 \left(\boldsymbol{W}_t + \frac{2 \log^4 T}{(1-\gamma)T}\mathbf{1}\right) \log \frac{8KT}{\delta}} + \frac{4}{3} B \log \frac{8KT \log \frac{1}{1-\gamma}}{\delta} \mathbf{1}
$$

$$
\leq \sqrt{\frac{32 \log^4 T}{(1-\gamma)T} \log \frac{8KT}{\delta} \left( \max_{(1-\alpha)t \leq i < t} \text{Var}_{\boldsymbol{\Lambda}\boldsymbol{P}_\mathcal{K}}\left(\boldsymbol{V}_i\right) + \mathbf{1} \right)} + \frac{12 \log^4 T}{(1-\gamma)^2 T} \log \frac{8KT}{\delta} \mathbf{1}. \tag{50}
$$

Hence if we define

$$
\boldsymbol{\varphi}_t := 64 \frac{\log^4 T \log \frac{KT}{\delta}}{(1-\gamma)T} \left( \max_{\frac{t}{2} \leq i \leq t} \text{Var}_{\overline{\boldsymbol{P}}}\left(\boldsymbol{V}_i\right) + \mathbf{1} \right),
$$

then (46) and (50) implies that

$$
|\boldsymbol{\theta}_t| + |\boldsymbol{\nu}_t| + |\boldsymbol{\omega}_t| \leq \sqrt{\boldsymbol{\varphi}_t} + \frac{2\gamma\xi}{1-\gamma}\mathbf{1}, \tag{51}
$$

with probability over $1-\delta$ for all $2t/3 \leq k \leq t$, as long as $T \gg \log^4 T \log \frac{KT}{\delta} / (1-\gamma)^3$. Therefore, plugging (51) into (45), we arrive at the recursive relationship

$$
\boldsymbol{\Delta}_t \leq \sqrt{\boldsymbol{\varphi}_t} + \frac{2\gamma\xi}{1-\gamma}\mathbf{1} + \sum_{i=(1-\alpha)k+1}^{k} \eta_i^{(k)} \gamma \overline{\boldsymbol{P}}^{\pi_{i-1}} \boldsymbol{\Delta}_{i-1} = \sqrt{\boldsymbol{\varphi}_t} + \frac{2\gamma\xi}{1-\gamma}\mathbf{1} + \sum_{i=(1-\alpha)k}^{k-1} \eta_i^{(k)} \gamma \overline{\boldsymbol{P}}^{\pi_{i-1}} \boldsymbol{\Delta}_i.
$$

This recursion is expressed in a similar way as [4, Eqn. (46)] so we can invoke similar derivation in [4, Appendix A.2] to obtain that

$$
\boldsymbol{\Delta}_t \leq 30 \sqrt{\frac{\log^4 T \log \frac{KT}{\delta}}{(1-\gamma)^4 T} \left(1 + \max_{\frac{t}{2} \leq i < t} \|\boldsymbol{\Delta}_i\|_\infty\right)} \mathbf{1} + \frac{2\gamma\xi}{(1-\gamma)^2}\mathbf{1}. \tag{52}
$$

Then we turn to (44). Applying a similar argument, we can deduce that

$$
\boldsymbol{\Delta}_t \geq -30 \sqrt{\frac{\log^4 T \log \frac{KT}{\delta}}{(1-\gamma)^4 T} \left(1 + \max_{\frac{t}{2} \leq i < t} \|\boldsymbol{\Delta}_i\|_\infty\right)} \mathbf{1} - \frac{2\gamma\xi}{(1-\gamma)^2}\mathbf{1}. \tag{53}
$$

For any $t$ satisfying $\frac{T}{c_2 \log \frac{1}{1-\gamma}} \leq t \leq T$, taking (52) and (53) collectively gives rise to

$$\|\boldsymbol{\Delta}_t\|_\infty \leq 30 \sqrt{\frac{\log^4 T \log \frac{KT}{\delta}}{(1-\gamma)^4 T} \left(1 + \max_{\frac{t}{2} \leq i < t} \|\boldsymbol{\Delta}_i\|_\infty\right)} + \frac{2\gamma\xi}{(1-\gamma)^2}. \tag{54}$$

Let

$$u_k := \max\left\{\|\boldsymbol{\Delta}_t\|_\infty : 2^k \frac{T}{c_2 \log \frac{1}{1-\gamma}} \leq t \leq T\right\}.$$

By taking supremum over $t \in \{\lceil 2^k T/(c_2 \log \frac{1}{1-\gamma})\rceil, \ldots, T\}$ on both sides of (54), we have

$$u_k \leq 30 \sqrt{\frac{\log^4 T \log \frac{KT}{\delta}}{(1-\gamma)^4 T} (1 + u_{k-1})} + \frac{2\gamma\xi}{(1-\gamma)^2} \qquad \forall\, 1 \leq k \leq \log\left(c_2 \log \frac{1}{1-\gamma}\right). \tag{55}$$

It is straightforward to bound $u_0 \leq \frac{1}{1-\gamma}$. For $k \geq 1$, it is straightforward to obtain from (55) that

$$u_k \leq 3\max\left\{30\sqrt{\frac{\log^4 T \log \frac{KT}{\delta}}{(1-\gamma)^4 T}}, 30\sqrt{\frac{\log^4 T \log \frac{KT}{\delta}}{(1-\gamma)^4 T} u_{k-1}}, \frac{2\gamma\xi}{(1-\gamma)^2}\right\}, \tag{56}$$

for $1 \leq k \leq \log(c_2 \log \frac{1}{1-\gamma})$. We analyze (56) under two different cases:

1. If there exists some integer $k_0$ with $1 \leq k_0 < \lceil\log(c_2 \log \frac{1}{1-\gamma})\rceil$, such that

$$u_{k_0} \leq \max\left\{1, \frac{6\gamma\xi}{(1-\gamma)^2}\right\},$$

then it is straightforward to check from (56) that

$$u_{k_0+1} \leq 3\max\left\{30\sqrt{\frac{\log^4 T \log \frac{KT}{\delta}}{(1-\gamma)^4 T}}, \frac{2\gamma\xi}{(1-\gamma)^2}\right\} \tag{57}$$

as long as $T \geq C_3(1-\gamma)^{-4} \log^4 T \log(KT/\delta)$ for some sufficiently large constant $C_3 > 0$.

2. Otherwise we have $u_k > \max\{1, \frac{6\gamma\xi}{(1-\gamma)^2}\}$ for all $1 \leq k < \lceil\log(c_2 \log \frac{1}{1-\gamma})\rceil$. This together with (56) suggests that

$$\max\left\{1, \frac{6\gamma\xi}{(1-\gamma)^2}\right\} < 3\max\left\{30\sqrt{\frac{\log^4 T \log \frac{KT}{\delta}}{(1-\gamma)^4 T}}, 30\sqrt{\frac{\log^4 T \log \frac{KT}{\delta}}{(1-\gamma)^4 T} u_{k-1}}, \frac{2\gamma\xi}{(1-\gamma)^2}\right\},$$

and therefore

$$\max\left\{30\sqrt{\frac{\log^4 T \log \frac{KT}{\delta}}{(1-\gamma)^4 T}}, 30\sqrt{\frac{\log^4 T \log \frac{KT}{\delta}}{(1-\gamma)^4 T} u_{k-1}}, \frac{2\gamma\xi}{(1-\gamma)^2}\right\} = 30\sqrt{\frac{\log^4 T \log \frac{KT}{\delta}}{(1-\gamma)^4 T} u_{k-1}}$$

for all $1 \leq k \leq \log(c_2 \log \frac{1}{1-\gamma})$. Let

$$v_k := 90\sqrt{\frac{\log^4 T \log \frac{KT}{\delta}}{(1-\gamma)^4 T} u_{k-1}}.$$

Then we know from (55) that

$$u_k \leq v_k \qquad \forall\, 1 \leq k \leq \log\left(c_2 \log \frac{1}{1-\gamma}\right).$$

By applying the above two inequalities recursively, we know that

$$u_k \le v_k = \left(\frac{8100 \log^4 T \log \frac{KT}{\delta}}{(1-\gamma)^4 T}\right)^{1/2} u_{k-1}^{1/2} \le \left(\frac{8100 \log^4 T \log \frac{KT}{\delta}}{(1-\gamma)^4 T}\right)^{1/2} v_{k-1}^{1/2}$$

$$\le \left(\frac{8100 \log^4 T \log \frac{KT}{\delta}}{(1-\gamma)^4 T}\right)^{1/2+1/4} u_{k-2}^{1/4} \le \left(\frac{8100 \log^4 T \log \frac{KT}{\delta}}{(1-\gamma)^4 T}\right)^{1/2+1/4} v_{k-2}^{1/4}$$

$$\le \cdots \le \left(\frac{8100 \log^4 T \log \frac{KT}{\delta}}{(1-\gamma)^4 T}\right)^{1-1/2^k} u_0^{1/2^k} \le \sqrt{\frac{8100 \log^4 T \log \frac{KT}{\delta}}{(1-\gamma)^4 T}} \left(\frac{1}{1-\gamma}\right)^{1/2^k},$$

where the last inequality holds as long as $T \ge C_3 \log^4 T \log(KT/\delta)(1-\gamma)^{-4}$ for some sufficiently large constant $C_3 > 0$. Let $k_0 = \widetilde{c} \log \log \frac{1}{1-\gamma}$ for some properly chosen constant $\widetilde{c} > 0$ such that $k_0$ is an integer between 1 and $\log(c_2 \log \frac{1}{1-\gamma})$, we have

$$u_{k_0} \le \sqrt{\frac{8100 \log^4 T \log \frac{KT}{\delta}}{(1-\gamma)^4 T}} \left(\frac{1}{1-\gamma}\right)^{1/2^{k_0}} = O\left(\sqrt{\frac{\log^4 T \log \frac{KT}{\delta}}{(1-\gamma)^4 T}}\right).$$

When $T \ge C_3 \log^4 T \log(KT/\delta)(1-\gamma)^{-4}$ for some sufficiently large constant $C_3 > 0$, this implies that $u_{k_0} < 1$, which contradicts with the preassumption that $u_k > \max\{1, \frac{6\gamma\xi}{(1-\gamma)^2}\}$ for all $1 \le k \le c_2 \log \frac{1}{1-\gamma}$.

Consequently, (57) must hold true and then the definition of $u_k$ immediately leads to

$$\|\boldsymbol{\Delta}_T\|_\infty \le 90\sqrt{\frac{\log^4 T \log \frac{KT}{\delta}}{(1-\gamma)^4 T}} + \frac{6\gamma\xi}{(1-\gamma)^2}.$$

Then for any $\varepsilon \in (0, 1]$, one has

$$\|\boldsymbol{\Delta}_T\|_\infty \le \varepsilon + \frac{6\gamma\xi}{(1-\gamma)^2},$$

as long as

$$90\sqrt{\frac{\log^4 T \log \frac{KT}{\delta}}{(1-\gamma)^4 T}} \le \varepsilon.$$

Hence, if the total number of iterations $T$ satisfies

$$T \ge C_3 \frac{\log^4 T \log \frac{KT}{\delta}}{(1-\gamma)^4 \varepsilon^2}$$

for some sufficiently large constant $C_3 > 0$, (10) would hold for Algorithm 1 with probability over $1 - \delta$.

Finally, we are left to justify (48). Recall the definition of $\boldsymbol{x}_i$ (cf. (47)), one has

$$\mathrm{diag}\left(\mathsf{Var}\left(\boldsymbol{x}_i | \boldsymbol{V}_{i-1}, \cdots, \boldsymbol{V}_0\right)\right) = \gamma^2 \left(\eta_i^{(t)}\right)^2 \mathrm{diag}\left(\mathsf{Var}\left(\boldsymbol{\Lambda}\left(\widehat{\boldsymbol{P}}_{\mathcal{K}}^{(t)} - \boldsymbol{P}_{\mathcal{K}}\right)\boldsymbol{V}_{i-1} | \boldsymbol{V}_{i-1}\right)\right)$$

$$= \gamma^2 \left(\eta_i^{(t)}\right)^2 \mathrm{diag}\left(\boldsymbol{\Lambda}\mathsf{Var}\left(\left(\widehat{\boldsymbol{P}}_{\mathcal{K}}^{(i)} - \boldsymbol{P}_{\mathcal{K}}\right)\boldsymbol{V}_{i-1} | \boldsymbol{V}_{i-1}\right)\boldsymbol{\Lambda}^\top\right)$$

$$= \gamma^2 \left(\eta_i^{(t)}\right)^2 \left\{\boldsymbol{\lambda}(s, a)^2 \mathsf{Var}_{\boldsymbol{P}_{\mathcal{K}}}\left(\boldsymbol{V}_{i-1}\right)\right\}_{s,a},$$

where the notation $\mathsf{Var}_{\boldsymbol{P}_{\mathcal{K}}}(\boldsymbol{V}_{i-1})$ is defined in (12). Plugging this into the definition of $\boldsymbol{W}_t$ leads to

$$\boldsymbol{W}_t = \gamma^2 \sum_{i=(1-\alpha)t+1}^{t} \left(\eta_i^{(t)}\right)^2 \left\{\boldsymbol{\lambda}(s, a)^2 \mathsf{Var}_{\boldsymbol{P}_{\mathcal{K}}}\left(\boldsymbol{V}_{i-1}\right)\right\}_{s,a}$$

$$= \gamma^2 \sum_{i=(1-\alpha)t+1}^{t} \left(\eta_i^{(t)}\right)^2 \left\{\boldsymbol{\lambda}(s, a)^2 \left(\boldsymbol{P}_{\mathcal{K}}\left(\boldsymbol{V}_{i-1} \circ \boldsymbol{V}_{i-1}\right) - \left(\boldsymbol{P}_{\mathcal{K}}\boldsymbol{V}_{i-1}\right) \circ \left(\boldsymbol{P}_{\mathcal{K}}\boldsymbol{V}_{i-1}\right)\right)\right\}_{s,a}. \quad (58)$$

Then we introduce a useful claim as follows. The proof is deferred to Appendix C.2.

*Claim* 1. For any state-action pair $(s, a) \in \mathcal{S} \times \mathcal{A}$ and vector $\boldsymbol{V} \in \mathbb{R}^{|\mathcal{S}|}$, one has

$$
\begin{aligned}
\boldsymbol{\lambda}\left(s, a\right)^{2} & \left(\boldsymbol{P}_{\mathcal{K}}\left(\boldsymbol{V} \circ \boldsymbol{V}\right) - \left(\boldsymbol{P}_{\mathcal{K}}\boldsymbol{V}\right) \circ \left(\boldsymbol{P}_{\mathcal{K}}\boldsymbol{V}\right)\right) \\
& \leq \boldsymbol{\lambda}\left(s, a\right)\boldsymbol{P}_{\mathcal{K}}\left(\boldsymbol{V} \circ \boldsymbol{V}\right) - \left(\boldsymbol{\lambda}\left(s, a\right)\boldsymbol{P}_{\mathcal{K}}\boldsymbol{V}\right) \circ \left(\boldsymbol{\lambda}\left(s, a\right)\boldsymbol{P}_{\mathcal{K}}\boldsymbol{V}\right).
\end{aligned} \tag{59}
$$

By invoking this claim with $\boldsymbol{V} = \boldsymbol{V}^{i-1}$ and taking collectively with (58), one has

$$
\begin{aligned}
\boldsymbol{W}_{t} & \leq \gamma^{2} \sum_{i=(1-\beta)t+1}^{t} \left(\eta_{i}^{(t)}\right)^{2} \left\{\boldsymbol{\lambda}\left(s, a\right)\boldsymbol{P}_{\mathcal{K}}\left(\boldsymbol{V}_{i-1} \circ \boldsymbol{V}_{i-1}\right) - \left(\boldsymbol{\lambda}\left(s, a\right)\boldsymbol{P}_{\mathcal{K}}\boldsymbol{V}_{i-1}\right) \circ \left(\boldsymbol{\lambda}\left(s, a\right)\boldsymbol{P}_{\mathcal{K}}\boldsymbol{V}_{i-1}\right)\right\}_{s,a} \\
& = \gamma^{2} \sum_{i=(1-\beta)t+1}^{t} \left(\eta_{i}^{(t)}\right)^{2} \left[\boldsymbol{\Lambda}\boldsymbol{P}_{\mathcal{K}}\left(\boldsymbol{V}_{i-1} \circ \boldsymbol{V}_{i-1}\right) - \left(\boldsymbol{\Lambda}\boldsymbol{P}_{\mathcal{K}}\boldsymbol{V}_{i-1}\right) \circ \left(\boldsymbol{\Lambda}\boldsymbol{P}_{\mathcal{K}}\boldsymbol{V}_{i-1}\right)\right] \\
& = \gamma^{2} \sum_{i=(1-\beta)t+1}^{t} \left(\eta_{i}^{(t)}\right)^{2} \mathsf{Var}_{\overline{\boldsymbol{P}}}\left(\boldsymbol{V}_{i-1}\right),
\end{aligned}
$$

which is the desired result.

## C.2 Proof of Claim 1

To simplify notations in this proof, we use $[\lambda_{i}]_{i=1}^{K}$, $[P_{i,j}]_{1 \leq i \leq K, 1 \leq j \leq |\mathcal{S}|}$ and $[V_{i}]_{i=1}^{|\mathcal{S}|}$ to denote $\boldsymbol{\lambda}(s, a)$, $\boldsymbol{P}_{\mathcal{K}}$ and $\boldsymbol{V}$ respectively. Then one has

$$
\begin{aligned}
\boldsymbol{\lambda}\left(s, a\right) & \boldsymbol{P}_{\mathcal{K}}\left(\boldsymbol{V} \circ \boldsymbol{V}\right) - \left(\boldsymbol{\lambda}\left(s, a\right)\boldsymbol{P}_{\mathcal{K}}\boldsymbol{V}\right) \circ \left(\boldsymbol{\lambda}\left(s, a\right)\boldsymbol{P}_{\mathcal{K}}\boldsymbol{V}\right) \\
& - \boldsymbol{\lambda}\left(s, a\right)^{2}\left(\boldsymbol{P}_{\mathcal{K}}\left(\boldsymbol{V} \circ \boldsymbol{V}\right) - \left(\boldsymbol{P}_{\mathcal{K}}\boldsymbol{V}\right) \circ \left(\boldsymbol{P}_{\mathcal{K}}\boldsymbol{V}\right)\right) \\
& = \sum_{i=1}^{K}\sum_{j=1}^{|\mathcal{S}|} \lambda_{i} P_{i,j} V_{j}^{2} - \left(\sum_{i=1}^{K}\sum_{j=1}^{|\mathcal{S}|} \lambda_{i} P_{i,j} V_{j}\right)^{2} - \sum_{i=1}^{K}\sum_{j=1}^{|\mathcal{S}|} \lambda_{i}^{2} P_{i,j} V_{j}^{2} + \sum_{i=1}^{K} \lambda_{i}^{2}\left(\sum_{j=1}^{|\mathcal{S}|} P_{i,j} V_{j}\right)^{2} \\
& = \sum_{i=1}^{K}\sum_{j=1}^{|\mathcal{S}|} \lambda_{i} P_{i,j} V_{j}\left[\left(1 - \lambda_{i}\right) V_{j} - \sum_{i' \neq i}\sum_{j'=1}^{|\mathcal{S}|} \lambda_{i'} P_{i',j'} V_{j'}\right]. \\
& = \sum_{i=1}^{K}\sum_{j=1}^{|\mathcal{S}|} \lambda_{i} P_{i,j} V_{j}\left[\left(\sum_{i'=1}^{K}\sum_{j'=1}^{|\mathcal{S}|} \lambda_{i'} P_{i',j'} - \lambda_{i}\right) V_{j} - \sum_{i' \neq i}\sum_{j'=1}^{|\mathcal{S}|} \lambda_{i'} P_{i',j'} V_{j'}\right] \\
& = \sum_{i=1}^{K}\sum_{j=1}^{|\mathcal{S}|}\sum_{i' \neq i}\sum_{j'=1}^{|\mathcal{S}|} \lambda_{i} P_{i,j} V_{j} \lambda_{i'} P_{i',j'}\left(V_{j} - V_{j'}\right)
\end{aligned}
$$

where in the penultimate equality, we use the fact that

$$
\sum_{i'=1}^{K}\sum_{j'=1}^{|\mathcal{S}|} \lambda_{i'} P_{i',j'} = \boldsymbol{\lambda}\left(s, a\right)\boldsymbol{P}_{\mathcal{K}}\boldsymbol{1} = 1.
$$

It follows that

$$
\boldsymbol{\lambda}\left(s, a\right) \boldsymbol{P}_{\mathcal{K}}\left(\boldsymbol{V} \circ \boldsymbol{V}\right) - \left(\boldsymbol{\lambda}\left(s, a\right) \boldsymbol{P}_{\mathcal{K}} \boldsymbol{V}\right) \circ \left(\boldsymbol{\lambda}\left(s, a\right) \boldsymbol{P}_{\mathcal{K}} \boldsymbol{V}\right)
$$
$$
- \boldsymbol{\lambda}\left(s, a\right)^2 \left(\boldsymbol{P}_{\mathcal{K}}\left(\boldsymbol{V} \circ \boldsymbol{V}\right) - \left(\boldsymbol{P}_{\mathcal{K}} \boldsymbol{V}\right) \circ \left(\boldsymbol{P}_{\mathcal{K}} \boldsymbol{V}\right)\right)
$$
$$
= \sum_{i=1}^{K} \sum_{1 \leq i' < i} \sum_{j=1}^{|\mathcal{S}|} \sum_{j'=1}^{|\mathcal{S}|} \left[\lambda_i P_{i,j} V_j \lambda_{i'} P_{i',j'}\left(V_j - V_{j'}\right) + \lambda_{i'} P_{i',j} V_j \lambda_i P_{i,j'}\left(V_j - V_{j'}\right)\right]
$$
$$
= \sum_{i=1}^{K} \sum_{1 \leq i' < i} \lambda_i \lambda_{i'} \left[\sum_{j=1}^{|\mathcal{S}|} \sum_{j'=1}^{|\mathcal{S}|} P_{i,j} V_j P_{i',j'}\left(V_j - V_{j'}\right) + \sum_{j=1}^{|\mathcal{S}|} \sum_{j'=1}^{|\mathcal{S}|} P_{i',j} V_j P_{i,j'}\left(V_j - V_{j'}\right)\right]
$$
$$
\overset{\text{(i)}}{=} \sum_{i=1}^{K} \sum_{1 \leq i' < i} \lambda_i \lambda_{i'} \left[\sum_{j=1}^{|\mathcal{S}|} \sum_{j'=1}^{|\mathcal{S}|} P_{i,j} V_j P_{i',j'}\left(V_j - V_{j'}\right) + \sum_{j=1}^{|\mathcal{S}|} \sum_{j'=1}^{|\mathcal{S}|} P_{i',j'} V_{j'} P_{i,j}\left(V_{j'} - V_j\right)\right]
$$
$$
= \sum_{i=1}^{K} \sum_{1 \leq i' < i} \lambda_i \lambda_{i'} \left[\sum_{j=1}^{|\mathcal{S}|} \sum_{j'=1}^{|\mathcal{S}|} P_{i,j} P_{i',j'}\left(V_j - V_{j'}\right)^2\right]
$$
$$
\geq 0,
$$

where in (i), we exchange the indices $j$ and $j'$.

## D   Feature dimension and the number of anchor state-action pairs

The assumption that the feature dimension (denoted by $K_{\mathsf{d}}$) and the number of anchor state-action pairs (denoted by $K_{\mathsf{n}}$) are equal is actually non-essential. In what follows, we will show that if $K_{\mathsf{d}} \neq K_{\mathsf{n}}$, then we can modify the current feature mapping $\phi : \mathcal{S} \times \mathcal{A} \to \mathbb{R}^{K_{\mathsf{d}}}$ to achieve a new feature mapping $\phi' : \mathcal{S} \times \mathcal{A} \to \mathbb{R}^{K_{\mathsf{n}}}$ that does not change the transition model $P$. By doing so, the new feature dimension $K_{\mathsf{n}}$ equals to the number of anchor state-action pairs.

To begin with, we recall from Definition 1 that there exists $K_{\mathsf{d}}$ unknown functions $\psi_1, \cdots, \psi_{K_{\mathsf{d}}} : \mathcal{S} \to \mathbb{R}$, such that

$$
P\left(s' | s, a\right) = \sum_{k=1}^{K_{\mathsf{d}}} \phi_k\left(s, a\right) \psi_k\left(s'\right),
$$

for every $(s, a) \in \mathcal{S} \times \mathcal{A}$ and $s' \in \mathcal{S}$. In addition, we also recall from Assumption 1 that there exists $\mathcal{K} \subseteq \mathcal{S} \times \mathcal{A}$ with $|\mathcal{K}| = K_{\mathsf{n}}$ such that for any $(s, a) \in \mathcal{S} \times \mathcal{A}$,

$$
\phi\left(s, a\right) = \sum_{i:(s_i, a_i) \in \mathcal{K}} \lambda_i\left(s, a\right) \phi\left(s_i, a_i\right) \in \mathbb{R}^{K_{\mathsf{d}}} \quad \text{for} \quad \sum_{i=1}^{K_{\mathsf{n}}} \lambda_i\left(s, a\right) = 1 \quad \text{and} \quad \lambda_i\left(s, a\right) \geq 0.
$$

**Case 1:** $K_{\mathsf{d}} > K_{\mathsf{n}}$. In this case, the vectors in $\{\phi(s, a) : (s, a) \in \mathcal{K}\}$ are linearly independent. For ease of presentation and without loss of generality, we assume that $K_{\mathsf{d}} = K_{\mathsf{n}} + 1$. This indicates that the matrix $\boldsymbol{\Phi} \in \mathbb{R}^{K_{\mathsf{d}} \times (|\mathcal{S}||\mathcal{A}|)}$ whose columns are composed of the feature vectors of all state-action pairs has rank $K_{\mathsf{n}}$ and is hence not full row rank. This suggests that there exists $K_{\mathsf{n}}$ linearly independent rows (without loss of generality, we assume they are the first $K_{\mathsf{n}}$ rows). We can remove the last row from $\boldsymbol{\Phi}$ to obtain $\boldsymbol{\Phi}' := \boldsymbol{\Phi}_{1:K_{\mathsf{n}},:} \in \mathbb{R}^{K_{\mathsf{n}} \times (|\mathcal{S}||\mathcal{A}|)}$ such that $\boldsymbol{\Phi}'$ is full row rank. Then we show that we can actually use the columns of $\boldsymbol{\Phi}'$ as new feature mappings. To see why this is true, note that the last row $\boldsymbol{\Phi}_{K_{\mathsf{n}}+1,:}$ can be represented as a linear combination of the first $K_{\mathsf{n}}$ rows, namely there must exist constants $\{c_k\}_{k=1}^{K_{\mathsf{n}}}$ such that for any $(s, a) \in \mathcal{S} \times \mathcal{A}$,

$$
\phi_{K_{\mathsf{n}}+1}(s, a) = \sum_{k=1}^{K_{\mathsf{n}}} c_k \phi_k(s, a).
$$

Define $\psi'_k = \psi_k + c_k \psi_{K_n + 1}$ for $k = 1, \ldots, K_n$, we have

$$P\left(s'|s,a\right) = \sum_{k=1}^{K_d} \phi_k\left(s,a\right)\psi_k\left(s'\right) = \phi_{K_n+1}\left(s,a\right)\psi_{K_n+1}\left(s'\right) + \sum_{k=1}^{K_n} \phi_k\left(s,a\right)\psi_k\left(s'\right)$$

$$= \sum_{k=1}^{K_n} \phi_k\left(s,a\right)\left[\psi_k\left(s'\right) + c_k\psi_{K_n+1}\left(s'\right)\right] = \sum_{k=1}^{K_n} \phi_k\left(s,a\right)\psi'_k\left(s'\right),$$

which is linear with respect to the new $K_n$ dimensional feature vectors. It is also straightforward to check that the new feature mapping satisfies Assumption 1 with the original anchor state-action pairs $\mathcal{K}$.

**Case 2:** $K_d < K_n$. For ease of presentation and without loss of generality, we assume that $K_n = K_d + 1$ and that the subspace spanned by the feature vectors of anchor state-action pairs is non-degenerate, i.e., has rank $K_d$ (otherwise we can use similar method as in Case 1 to further reduce the feature dimension $K_d$). In this case, the matrix $\mathbf{\Phi}_{\mathcal{K}} \in \mathbb{R}^{K_d \times K_n}$ whose columns are composed of the feature vectors of anchor state-action pairs has rank $K_d$. We can add $K_n - K_d = 1$ new row to $\mathbf{\Phi}_{\mathcal{K}}$ to obtain $\mathbf{\Phi}'_{\mathcal{K}} \in \mathbb{R}^{K_n \times K_n}$ such that $\mathbf{\Phi}'_{\mathcal{K}}$ has full rank $K_n$. Then we let the columns of $\mathbf{\Phi}'_{\mathcal{K}} = [\phi'(s,a)]_{(s,a) \in \mathcal{K}}$ to be the new feature vectors of the anchor state-action pairs, and define the new feature vectors for all other state-action pairs $(s,a) \notin \mathcal{K}$ by

$$\phi'\left(s,a\right) = \sum_{i:(s_i,a_i) \in \mathcal{K}} \lambda_i\left(s,a\right)\phi'\left(s_i,a_i\right).$$

We can check that the transition model $P$ is not changed if we let $\psi_{K_n}(s') = 0$ for every $s' \in \mathcal{S}$. It is also straightforward to check that Assumption 1 is satisfied.

To conclude, when $K_d \neq K_n$, we can always construct a new set of feature mappings with dimension $K_n$ such that: (i) the feature dimension equals to the number of anchor state-action pairs (they are both $K_n$); (ii) the transition model can still be linearly parameterized by this new set of feature mappings; and (iii) the anchor state-action pair assumption (Assumption 1) is satisfied with the original anchor state-action pairs.