# OpenReview forum: "Sample-Efficient Reinforcement Learning for Linearly-Parameterized MDPs with a Generative Model"
_NeurIPS.cc/2021/Conference — NeurIPS 2021 Poster_

### Official Review · Reviewer_G5iQ · 2021-07-04

**Rating:** 6
**Confidence:** 3

**Summary:**

This paper considers reinforcement learning in infinite horizon discounted Markov decision processes with access to a generative model with linearly parameterized transitions.  At a high level, the authors consider an infinite horizon discounted Markov decision process with discount factor $\gamma$, finite state and action spaces, and known reward functions.  The high level goal is to query the transition kernel at various states in order to learn a policy $\pi$ mapping states $s$ to actions $a$ which maximizes the cumulative reward experienced across the trajectory.

It is widely known that learning in general depends on $|S| \times |A|$ which can be prohibitively large whenever $S$ or $A$ is large.  As such, the authors impose additional assumptions in the paper different from prior work with respect to the transition kernel (the only uncertainty present in the model) in order to avoid this worst-case scaling.  In particular, they assume that the algorithm designer has access to a feature function $\phi : S \times A \rightarrow R^K$ such that there are unknown functions $\psi_1, \ldots, \psi_k$ where $P(s' | s,a) = \sum_k \phi_k(s,a) \psi_k(s').$  This definition is the same structure imposed on the probability transition kernel in the linear MDP model (but doesn't also require that the reward function is a linear mapping of the features $\phi$).  This model encompasses tabular MDPs with no additional assumptions where $K$ can be taken to be $S \times A$.

In addition to the linear assumption, the authors assume access to a set of anchor state action pairs of size $K$.  With this assumption, they assume that the feature vectors can be expressed as a convex combination of the feature vectors of the anchor state action pairs (which additionally imposes that the transition kernel can be written as $P(s' | s,a) = \sum_i \lambda_i(s,a) P(s' | s_i, a_i))$ where $i$ indexes over the anchor state action pairs. The authors assume that the $\lambda$ function and the anchor pairs are known in advance to the algorithm designer.

This assumption differentiates the model from the typical infinite horizon discounted MDP to (hopefully) obtain improved regret guarantees.  In particular, the theoretical contributions for the authors follows via two separate strands.

First, the authors consider a model based reinforcement learning algorithm using the generative model.  In particular, their algorithm queries all of the anchor state-action pairs a given number of times to obtain independent samples from the transition $P( | s_i, a_i)$.  Using this, they can then construct an empirical transition kernel for all state action pairs using the linear combination arising from the anchor state action pair assumption.  With this estimate of the transition kernel, the authors can construct an empirical MDP where the true transition is replaced by its estimate (which can be done as the reward function is known), and use planning to find an optimal policy in this perturbed MDP.

Via this method the authors show that the sample complexity scales via $K / (1 - \gamma)^3 \epsilon^2$, which is minimax optimal up to logarithmic factors.

Second, the authors consider a model free reinforcement learning algorithm using the generative model.  This algorithm proceeds across time steps, where in each time step they sample a transition for each anchor state action pair, and update an estimate for the $Q$ function using the 'anchored' Bellman equations.

Via this method the authors show that the sample complexity scales via $K / (1 - \gamma)^4 \epsilon^2$, which is minimax optimal or 'sharp' under naive $Q$ learning based algorithms.

Lastly, the authors complement the theoretical results by a detailed discussion comparing the theoretical guarantees on the performance of their approaches to that in [61].

**Ethical Concerns:**

The work presented in this paper considers a theoretical model for solving reinforcement learning problems via learning in an unknown infinite horizon Markov decision process. As such, there are little ethical considerations for this paper.


**Limitations And Societal Impact:**

The paper focuses on the theory of reinforcement learning, and so it is expected that it will not have any potential negative societal impacts.  The authors are also clear on the limitations of their work, including:
- Assumption that $S$ and $A$ are finite
- Accommodating a large range on $\epsilon$ for the sample complexity guarantees

In addition, there are two assumptions which are stated implicitly but the authors never explicitly discussed their importance:
- Known feature representation $\phi$ and anchor states
- Known reward function

**Main Review:**

### Originality:
The authors present an extension of existing model based and $Q$ learning based algorithms to the infinite horizon discounted MDP setting with linearly parameterized transition kernels.  While the algorithms presented are simple extensions of vanilla model based or $Q$ learning based algorithms, the authors express the necessary and straight-forward modifications to better exploit the linearly parameterized transitions in the algorithm design.  The proof techniques employed seemed like simple extensions of existing proof techniques, but see the questions section for more detailed comments.

### Quality:
The submission is technically sound and the theoretical claims are well supported.  The authors are honest and upfront on the new techniques used in both their proof and algorithm development.  However, there were certain additional assumptions required for the algorithm, namely:
- Known reward function
- Known feature representations $\phi$ and anchor states
which are never fully discussed or compared to the previous literature.

### Clarity:
The submission is clearly written and well organized.  However, with this there are a couple of confusing sections and theorem presentations:
- The title of the paper is misleading, as the authors do not consider the true 'linear MDP' model but instead 'linearly parameterized transitions'
- The sample complexity bounds for Theorem 1 are performance guarantees with respect to the $Q$ function of the learned policy.  However, the guarantees in Theorem 2 are just guarantees with respect to the estimated $Q$ function (independent of the performance of the greedy policy with respect to the estimated $Q$ function).  This is later discussed in the comments, but might be easier to understand if the theorems are in the same 'format'.  Also, upon checking the cited result to transform the bound, it does not provide performance guarantee of the greedy w/r/t $Q$ estimates in the true MDP as cited - unless I am misunderstanding the result.
- The key algorithmic differences between this paper and [61] are never fully discussed.
- Line 324 minor typo in 'employing more complicated analysis tool'.
- The relation to the other linear MDP models listed in line 286 is never fully discussed with respect to the different proof techniques and algorithmic tools needed.

### Significance:
The results in this paper improve the best known bounds for sample complexity guarantees with their model. While the novel techniques used are quite limited (just modifying existing generative model RL algorithms to exploit the anchor state representations), the work can be built upon by better understanding the required assumptions (finite state action space and known anchor state representation).

### Strengths:
The main strengths of the paper are as follows:
- The authors present minimax sample complexity bounds for the infinite horizon MDP setting with linear transitions and access to a generative model
- The paper is incredibly well written and easy to follow

### Weaknesses:
The main weaknesses of the paper are as follows:
- The novel contributions seem limited as the algorithms are straightforward extensions of existing generative model algorithms.  While it seems like the theoretical tools required are novel, these contributions were not fully highlighted in the paper.
- There are no concrete or realistic problem set-ups presented which satisfy the authors linearly parametrized MDP assumptions

### Questions and Comments:
- Line 118 - 121 was a nice point and helped highlight the improvements from the additional assumption.
- Line 134 - 137 was a useful discussion on the notion of anchor states.  However, the assumption that these are known as well as $\lambda(s,a)$ was never explicitly stated but required in the algorithms.  While it seems straightforward to learn the anchor state and $\lambda$ function just from the feature mapping $\phi$, this should be explicitly discussed in the paper.  Especially in the algorithm description you state the feature mapping $\phi$ only is needed (as in Algorithm 1).
- In Theorem 1, the notation $\epsilon_{opt}$ was introduced in the result but not in the theorem statement.  While there was a discussion after the theorem statement, this should be included in the formal theorem definition.
- Section on line 175 was a nice point to consider the computational complexity required.
- For the first point in section 5.  It seems a straightforward proof technique would be to show that the constructed estimated transition kernel approximates the true one up to some accuracy, and then use sensitivity of planning in the perturbed MDP with respect to the estimated transition kernel.  Did you try this approach and show that it leads to suboptimal guarantees?


EDIT: I'd like to thank the authors for their detailed comments for each of the concerns that they bring up, and I've updated my score accordingly.  The authors seem to have taken great care to improve the readability and clarity of the paper, especially addressing some of my earlier concerns with respect to constructing the set of anchor states from the feature representation $\phi$, and comparing some aspects of their proof techniques to other more 'naive' approaches to solving the problem.  All in all, my only concerns that remain are with respect to:
1. The usefulness of the model.  Even in responding to my comment the authors are unable to describe a concrete model where Assumption 1 is valid.  While it is still useful as a starting point for other theoretical analysis, having an example would help.
2. The strength of assumption 1.

**Time Spent Reviewing:**

4 hours

---

> ### Author Response · Authors · 2021-08-10
> **Response to Reviewer G5iQ (Part 1)**
>
> We thank gratefully the reviewer for careful reading of our manuscript
> and for very helpful comments and suggestions. We have tried our best
> to address the comments as follows.
>
> $\mathbf{Quality:}$
>
> 1. Known reward function
>
> $\mathsf{Response:}$ Thank you for raising this point. By checking the
> algorithms, we can see that this assumption is only needed in Algorithm
> 1 (i.e., the model-based method). Although assuming the reward function
> to be known is restrictive to some extent, it is widely imposed by
> prior literature on model-based RL [2,5,6,17,35]. Alternatively,
> if we do not assume the reward function to be known in Algorithm 1,
> we can instead assume that it is also linear with respect to the feature
> mapping $\phi$ (i.e. $r(s,a)=\theta^{\top}\phi(s,a),\quad\forall(s,a)\in\mathcal{S}\times\mathcal{A}$)
> [26,30,54,58], and then we can easily calculate $\theta$ by solving
> the following system of linear equations
> \begin{equation}
> \qquad\qquad\qquad\qquad\qquad\qquad\qquad\qquad\qquad
> 	r(s,a)=\theta^{\top}\phi(s,a),\quad\forall(s,a)\in\mathcal{K}.
> \end{equation}
> We have added detailed discussion about this in our revised paper.
>
> $\mathsf{Revised\ text:}$ "In Algorithm 1, the reward function $r$
> is assumed to be known. If the information of $r$ is unavailable
> to us, an alternative is to assume that $r$ is linear with respect
> to the feature mapping $\phi$, i.e. $r(s,a)=\theta^{\top}\phi(s,a)$
> for every $(s,a)\in\mathcal{S}\times\mathcal{A}$, which is widely
> adopted in linear MDP literature [26,30,54,58]. Under this linear
> assumption, one can obtain $\theta$ by solving the following linear
> system of equations
> \begin{equation}
> \qquad\qquad\qquad\qquad\qquad\qquad\qquad\qquad\qquad
> r\left(s,a\right)=\theta^{\top}\phi\left(s,a\right),\quad\forall\left(s,a\right)\in\mathcal{K},
> \end{equation}
> which can be constructed by the observed reward $r(s,a)$ for all
> anchor state-action pairs."
>
> 2. Known feature representations $\phi$ and anchor states which are
> never fully discussed or compared to the previous literature
>
> $\mathsf{Response:}$ Thank you for raising this. Regarding the feature
> mapping $\phi$, in practice, large scale RL usually makes use of
> representation learning to obtain the feature mapping $\phi$ [Bengio
> et al., 2013]. Furthermore, the learned representations can be selected
> to satisfy the anchor state-action pairs assumption by design. With
> the feature mapping $\phi$ in hand, the anchor state-action pairs
> $\mathcal{K}$ can actually be calculated by various algorithms as
> discussed in detail in Section 3.1 of [Jin et al., 2017]. Since
> this is not the focus of our paper, $\mathcal{K}$ is assumed to be
> known to us. We have added the following discussion in the revised
> paper.
>
> $\mathsf{Revised\ text:}$ "It is assumed that the feature mapping $\phi$
> is known throughout this paper, which is a widely adopted assumption
> in previous literature [61,30,65,54,58,50,26]. In practice, large
> scale RL usually makes use of representation learning to obtain the
> feature mapping $\phi$. Furthermore, the learned representations
> can be selected to satisfy the anchor state-action pairs assumption
> by design."
>
> $\mathbf{Clarity:}$
>
> 1. The title of the paper is misleading, as the authors do not consider
> the true 'linear MDP' model but instead 'linearly parameterized transitions'
>
> $\mathsf{Response:}$ Thank you for the comment. Compared with linear
> MDP in [30], our assumption on the probability transition matrix
> $P$ is the same as theirs, while we assume that the reward function
> $r$ is known rather than linear with respect to the feature mapping
> $\phi$ as in [30]. The assumption on the reward function was
> discussed in detail in our response to your first comment in Quality
> above. In the title, we use the term ``Linearly-Parameterized MDP''
> to differentiate from linear MDP in prior literature and its precise
> meaning is made clear in Definition 1 and the comment therein. We
> are happy to discuss with the reviewer if a new title is necessary
> in the later reviewing process.
>
> 2. The sample complexity bounds for Theorem 1 are performance guarantees
> with respect to the $Q$ function of the learned policy. However,
> the guarantees in Theorem 2 are just guarantees with respect to the
> estimated $Q$ function (independent of the performance of the greedy
> policy with respect to the estimated $Q$ function). This is later
> discussed in the comments, but might be easier to understand if the
> theorems are in the same 'format'. Also, upon checking the cited result
> to transform the bound, it does not provide performance guarantee
> of the greedy w/r/t $Q$ estimates in the true MDP as cited - unless
> I am misunderstanding the result.
>
> $\mathsf{Response:}$ Thank you for pointing out this confusion! We have
> added the performance guarantee of the learned policy to Theorem 2
> in the revised paper as below. In addition, when using [47] to
> transform the results, we first invoke [47] to obtain $\Vert V^{\pi_{T}}-V^{\star}\Vert_{\infty}\leq\frac{2\gamma\Vert V_{T}-V^{\star}\Vert_{\infty}}{1-\gamma}$
> ($\pi_{T}$ (resp. $V_{T}$) is the policy (resp. value function)
> induced by $Q_{T}$) and then due to $\Vert V_{T}-V^{\star}\Vert_{\infty}\leq\Vert Q_{T}-Q^{\star}\Vert_{\infty}$,
> we have $\Vert V^{\pi_{T}}-V^{\star}\Vert_{\infty}\leq\frac{2\gamma\Vert Q_{T}-Q^{\star}\Vert_{\infty}}{1-\gamma}$.
> This leads to the bound $\widetilde{O}(\frac{K}{(1-\gamma)^{6}\varepsilon^{2}})$
> in the comments. We have added this missing step in the proof of
> the revised theorem in the supplementary materials.
>
> $\mathsf{Revised\ text:}$ "In addition, let $\pi_{T}$ (resp. $V_{T}$)
> to be the policy (resp. value function) induced by $Q_{T}$, then
> one has
> \begin{equation}
> \qquad\qquad\qquad\qquad\qquad\qquad\qquad\qquad\qquad
> 	\max_{s\in\mathcal{S}}\left|V^{\pi_{T}}\left(s\right)-V^{\star}\left(s\right)\right|\leq\frac{2\gamma\varepsilon}{1-\gamma}.
> \end{equation}"
>
> 3. The key algorithmic differences between this paper and [61]
> are never fully discussed.
>
> $\mathsf{Response:}$ Thank you for raising this point! Since [61]
> focuses on model-free Q-learning, we only need to compare Algorithm
> 2 with [61]. There are two algorithms proposed by [61]: Phased
> Parametric Q-Learning and Optimal Phased Parametric Q-Learning. Compared
> with the first one, Algorithm 2 in this paper maintains and updates
> a $Q$-function estimate $Q_{t}$, while Phased Parametric Q-Learning
> parameterized $Q$-function by
> \begin{equation}
> \qquad\qquad\qquad\qquad\qquad\qquad\qquad\qquad\qquad
> Q_{w}\left(s,a\right):= r\left(s,a\right)+\gamma\phi\left(s,a\right)^{\top}w,
> \end{equation}
> and then it updates the parameters $w$. The second one in [61]
> is a variant of Q-learning with variance reduction, which is beyond
> the scope of the current paper on vanilla Q-learning. We have added
> the following discussion in the comparison with [61] below Theorem
> 2 in our revised paper.
>
> $\mathsf{Revised\ text:}$ "It is worth mentioning that the critical
> difference between Algorithm 1 and Phased Parametric Q-Learning in
> [61] is that Algorithm 1 maintains and updates a $Q$-function
> estimate $Q_{t}$, while [61] parameterized $Q$-function by
> \begin{equation}
> \qquad\qquad\qquad\qquad\qquad\qquad\qquad\qquad\qquad
> 	Q_{w}\left(s,a\right):= r\left(s,a\right)+\gamma\phi\left(s,a\right)^{\top}w,
> \end{equation}
> and then updates the parameters $w$. "
>
> 4. Line 324 minor typo in 'employing more complicated analysis tool'.
>
> $\mathsf{Response:}$ Thank you for pointing out the typo! We have corrected
> it in our revised paper.
>
> 5. The relation to the other linear MDP models listed in line 286
> is never fully discussed with respect to the different proof techniques
> and algorithmic tools needed.
>
> $\mathsf{Response:}$ Thank you for your comment. In terms of proof techniques,
> we have revised Section 5 and discussed the connection of proof techniques
> between our paper and the most relevant previous works in detail as
> will be shown shortly in our response to your first comment in Weaknesses
> below. Regarding algorithmic tools, we have added detailed comparisons
> between our paper and some selected previous papers in terms of algorithmic
> tools in the revised paper as follows.
>
> $\mathsf{Revised\ text:}$ "Among them, [61] studied linear transition
> model and provided tight sample complexity bounds for a new variant
> of Q-learning with the help of variance reduction. [30] focused
> on linear MDP and designed an algorithm called ``Least-Squares Value
> Iteration with UCB'' with both polynomial runtime and polynomial
> sample complexity without accessing generative model. [54] extended
> the study of linear MDP to the framework of reward-free reinforcement
> learning. [65] considered a different feature mapping called linear
> kernel MDP and devised an algorithm with polynomial regret bound without
> generative model."
>
> Additional References
>
> [Bengio et al., 2013] Representation learning: A review and new
> perspectives. IEEE transactions on pattern analysis and machine intelligence,
> 35(8), 1798-1828.
>
> [Jin et al., 2017] Estimating Network Memberships by Simplex Vertex
> Hunting

---

> ### Author Response · Authors · 2021-08-10
> **Response to Reviewer G5iQ (Part 2)**
>
> Here we continue our response to Reviewer G5iQ, due to character limitation in each reply.
>
> $\mathbf{Weaknesses:}$
>
> 1. The novel contributions seem limited as the algorithms are straightforward
> extensions of existing generative model algorithms. While it seems
> like the theoretical tools required are novel, these contributions
> were not fully highlighted in the paper.
>
> $\mathsf{Response:}$ Thanks for raising this point. Regarding [5],
> though it also studies model-based RL, our model setting and major
> technical approaches as detailed in Section 5 bear little resemblance
> to those in [5], and results in [5] have been improved by
> [2]. There are two papers [2,34] that are the most relevant
> to our paper. In what follows, we will compare our theoretical guarantees
> for model-based RL with [2] and also compare our theoretical guarantees
> for vanilla Q-learning with [34]. To begin with, it is worth noting
> that the model-based algorithm in our paper is quite different from
> that in [2] since we only need to sample for the anchor state-action
> pairs rather than every state-action pair, which enables significant
> improvement of sample complexity. In terms of techniques, the only
> similarity between these two pieces of work is that we share the same
> high level idea of leave-one-out MDP, while how to employ this idea
> in linear transition model has been unexplored before this paper.
> Theorem 1 in this paper generalizes the idea to linear transition
> model and accommodates model misspecification, which actually needs
> a lot of efforts since it calls for delicate and careful control of
> the error, as detailed in the proof in Appendix B. Moving forward,
> [34] studied vanilla Q-learning in the tabular MDP setting and
> also adopted Freedman's inequality. However, we would like to argue
> that it requires significant efforts in order to study linear transition
> model and also allow for model misspecification in the current paper.
> In Section C in the supplementary materials, it can be seen that to
> analyze the linear transition model, we encounter a series of technical
> challenges, like how to solve the recursive relationship (53) (whose
> solution is drastically different from [34] as we allow for model
> misspecification) and the establishment of claim (47), which is beyond
> the scope of [34]. Hence, we think the proof techniques, though
> motivated by and has has connections with previous work to some extent,
> has much novelty in the sense that the linearly-parameterized MDP
> requires more sophisticated analysis. We have emphasized the technical
> challenges in our revised paper as follows.
>
> $\mathsf{Revised\ text:}$ "For the model-based approach, we employ
> the leave-one-out analysis to
> decouple the complicated statistical dependency between the empirical
> probability transition model $\widehat{P}$ and the corresponding
> optimal policy. Specifically, [2] proposed to construct a collection
> of auxiliary MDPs where each one of them leaves out a single state
> $s$ by setting $s$ to be an absorbing state and keeping everything
> else untouched. We tailor this high level idea to the needs of linear
> transition model, then the independence between the newly constructed
> MDP with absorbing state $s$ and data samples collected at state
> $s$ will facilitate our analysis, as detailed in Lemma 1. Compared
> with [2], Theorem 1 extends the tabular setting studied in [2]
> to the linear transition model and accommodates model misspecification,
> which actually needs significant efforts as detailed in the supplementary
> materials. This leave-one-out type of analysis has been utilized in studying numerous problems by
> a long line of work, such as [22, 37, 51, 13, 12, 14, 15], just
> to name a few.
>
> To obtain tighter sample complexity bound than the previous one $\widetilde{O}(\frac{K}{(1-\gamma)^{7}\varepsilon^{2}})$
> in [61] for vanilla Q-learning, we invoke Freedman's
> inequality [24] for the concentration of an error term with martingale
> structure as illustrated in Section C in the supplementary materials,
> while the classical ones used in analyzing Q-learning are Hoeffding's
> inequality and Bernstein's inequality [61]. The
> use of Freedman's inequality helps us establish a recursive
> relation on $\\{\Vert Q_{t}-Q^{\star}\Vert_{\infty}\\}_{t=0}^{T}$,
> which consequently leads to the performance guarantee (9). It is worth
> mentioning that [34] also studied vanilla Q-learning in the tabular
> MDP setting and adopted Freedman's inequality, while we emphasize
> that it requires a lot of efforts and more delicate analyses in order
> to study linear transition model and also allow for model misspecification
> in the current paper, as detailed in the supplementary materials.
> "
>
> 2. There are no concrete or realistic problem set-ups presented which
> satisfy the authors linearly parameterized MDP assumptions.
>
> $\mathsf{Response:}$ Thank you for your comment! We agree that the assumption
> of linearly-parameterized MDP model is strong and its applications
> are quite limited in practice. We hope to study how to generalize
> and weaken this assumption in future work. Nevertheless, we feel that
> the current paper is meaningful in its present form. Linear assumption
> is perhaps the most simple and fundamental modeling assumption, and
> has been widely used in fields like statistics and dynamical system.
> Linear MDP has become increasingly popular in the theoretical study
> of RL in recent years since it enjoys great mathematical tractability.
> We hope that this manuscript can serve as a starting point for understanding
> more complicated settings in future works. In addition, our results
> allow model misspecification as shown in the comments below Theorem
> 1 and 2. This implies that our results can be applied as long as there
> exists a linear transition model close to the true transition model
> instead of requiring the true transition model to be exactly linear,
> which is much less restrictive.
>
> $\mathbf{Questions\ and\ Comments:}$
>
> 1. Line 134 - 137 was a useful discussion on the notion of anchor
> states. However, the assumption that these are known as well as $\lambda(s,a)$
> was never explicitly stated but required in the algorithms. While
> it seems straightforward to learn the anchor state and $\lambda$
> function just from the feature mapping $\phi$, this should be explicitly
> discussed in the paper. Especially in the algorithm description you
> state the feature mapping $\phi$ only is needed (as in Algorithm 1).
>
> $\mathsf{Response:}$ Thanks for pointing out this confusion. In our
> revised paper, we have emphasized that $\{\lambda(s,a):(s,a)\in\mathcal{S}\times\mathcal{A}\}$
> need to be learnt from the feature mappings $\phi$ and the set $\mathcal{K}$
> as the next paragraph shows. In addition, as you said, the set of
> anchor state-action pairs $\mathcal{K}$ can be calculated from $\phi$,
> which has been discussed in detail in our response to your second
> comment of Quality above.
>
> $\mathsf{Revised\ text:}$ "Careful readers may note that in Algorithm
> 1, $\{\lambda(s,a):(s,a)\in\mathcal{S}\times\mathcal{A}\}$ is used
> in the construction of $\widehat{P}$, while $\{\lambda(s,a):(s,a)\in\mathcal{S}\times\mathcal{A}\}$
> is not input into the algorithm. This is because given $\mathcal{K}$
> and $\phi$ are known, $\{\lambda(s,a):(s,a)\in\mathcal{S}\times\mathcal{A}\}$
> can be calculated explicitly. "
>
> 2. In Theorem 1, the notation $\varepsilon_{\mathsf{opt}}$ was introduced
> in the result but not in the theorem statement. While there was a
> discussion after the theorem statement, this should be included in
> the formal theorem definition.
>
> $\mathsf{Response:}$ Thank you for raising this! The target algorithm
> error level $\varepsilon_{\mathsf{opt}}$ is originally defined as
> the input of Algorithm 1, whose output is studied by Theorem 1. We
> will define $\varepsilon_{\mathsf{opt}}$ in the theorem in our revised
> paper.
>
> 3. For the first point in section 5. It seems a straightforward proof
> technique would be to show that the constructed estimated transition
> kernel approximates the true one up to some accuracy, and then use
> sensitivity of planning in the perturbed MDP with respect to the estimated
> transition kernel. Did you try this approach and show that it leads
> to suboptimal guarantees?
>
> $\mathsf{Response:}$ Thank you for the question. To the best of our
> knowledge, study of sensitivity of planning in the empirical MDP can
> not produce approximations accurate enough for us to deduce tight
> sample complexity bounds as this paper does. In deriving Theorem 1,
> our goal is to compare $Q^{\widehat{\pi}}$ and $Q^{\star}$. As (18)
> in the supplementary materials shows, the critical challenge is the
> dependency between the estimated probability transition model and
> the corresponding optimal policy, which precludes us from using standard
> concentration inequalities. As the first point in Section 5 explains,
> the main benefits from applying the ''leave-one-out'' analysis is
> to decouple the dependencies. It is unclear to us how to solve or
> avoid this challenge by using sensitivity of planning in the the perturbation
> MDP.

---

> > ### Comment · Reviewer_G5iQ · 2021-08-21
> > **Discussion**
> >
> > I'd like to thank you for your detailed comments for each of the concerns, and I've updated my score and edited the main review accordingly.

---

> > > ### Author Response · Authors · 2021-08-24
> > > **Discussion**
> > >
> > > Thank you for your response and update! We fully understand your remaining concerns. Regarding the linear MDP assumption, to the best of our knowledge, there are few practical real world problems satisfying this assumption. One of the most relevant concrete examples proposed in previous literature is the mountain car problem [1], which will be described briefly in our revised paper. Indeed, we can see the linear transition model enjoys great mathematical tractability, while exhibits limited ability to capture complicated models in practice. Hence just as detailed in our previous review, the major contribution of this paper is to provide intellectual understanding.
> > >
> > > [1] Carvalho, Diogo, Francisco S. Melo, and Pedro Santos. "A new convergent variant of Q-learning with linear function approximation." Advances in Neural Information Processing Systems 33 (2020): 19412-19421.

---

### Official Review · Reviewer_u4X6 · 2021-07-11

**Rating:** 6
**Confidence:** 4

**Summary:**

This paper derives sample complexity for both model-based RL algorithms and model-free RL algorithm (i.e. $Q$-learning) in the case where the underlying MDP admits a linear structure. The resulting sample complexity bounds in both cases are shown to be tight.

**Limitations And Societal Impact:**

No Societal Impact

**Main Review:**

The linear parametrized MDP model, although was studied in related literature, seems to be relatively restrictive. It would be nice if such model is motivated by some real world applications.

The contributions of this paper are purely theoretical. The two major results are the sample complexity bounds for model-based and model-free RL algorithms. It seems that the approach for obtaining the sample complexity bound of model-based algorithm is similar to [2,5], and the approach for obtaining the sample complexity bound of model-free $Q$-learning is similar to [34]. It should be clearly stated what the technical challenges are in extending the results of [2,5,34] to the setting of this paper.

The $Q$-learning algorithm studied in this paper uses synchronous update, while practically $Q$-learning updates $Q_k$ in an asynchronous manner based a trajectory of samples obtained from a behavioral policy.






**Time Spent Reviewing:**

1.5

---

> ### Author Response · Authors · 2021-08-10
> **Response to Reviewer u4X6**
>
> We thank gratefully the reviewer for careful reading of our manuscript
> and for very helpful comments and suggestions. We have tried our best
> to address the comments as follows.
>
> 1. The linear parameterized MDP model, although was studied in related
> literature, seems to be relatively restrictive. It would be nice if
> such model is motivated by some real world applications.
>
> $\mathsf{Response:}$ Thank you for your comment! We agree that the assumption
> of linearly-parameterized MDP model is strong and its applications
> are quite limited in practice. We hope to study how to generalize
> and weaken this assumption in future work. Nevertheless, we feel that
> the current paper is meaningful in its present form. Linear assumption
> is perhaps the most simple and fundamental modeling assumption, and
> has been widely used in fields like statistics and dynamical system.
> Linear MDP has become increasingly popular in the theoretical study
> of RL in recent years since it enjoys great mathematical tractability.
> We hope that this manuscript can serve as a starting point for understanding
> more complicated settings in future works. In addition, our results
> allow model misspecification as shown in the comments below Theorem
> 1 and 2. This implies that our results can be applied as long as there
> exists a linear transition model close to the true transition model
> (instead of requiring the true transition model to be exactly linear),
> which is much less restrictive.
>
> 2. The contributions of this paper are purely theoretical. The two
> major results are the sample complexity bounds for model-based and
> model-free RL algorithms. It seems that the approach for obtaining
> the sample complexity bound of model-based algorithm is similar to
> [2,5], and the approach for obtaining the sample complexity bound
> of model-free $Q$-learning is similar to [34]. It should be clearly
> stated what the technical challenges are in extending the results
> of [2,5,34] to the setting of this paper.
>
> $\mathsf{Response:}$ Thanks for raising this point. Regarding [5],
> though it also studies model-based RL, our model setting and major
> technical approaches as detailed in Section 5 bear little resemblance
> to those in [5], and results in [5] have been improved by
> [2]. There are two papers [2,34] that are the most relevant
> to our paper. In what follows, we will compare our theoretical guarantees
> for model-based RL with [2] and also compare our theoretical guarantees
> for vanilla Q-learning with [34]. To begin with, it is worth noting
> that the model-based algorithm in our paper is quite different from
> that in [2] since we only need to sample for the anchor state-action
> pairs rather than every state-action pair, which enables significant
> improvement of sample complexity. In terms of techniques, the only
> similarity between these two pieces of work is that we share the same
> high level idea of leave-one-out MDP, while how to employ this idea
> in linear transition model has been unexplored before this paper.
> Theorem 1 in this paper generalizes the idea to linear transition
> model and accommodates model misspecification, which actually needs
> a lot of efforts since it calls for delicate and careful control of
> the error, as detailed in the proof in Appendix B. Moving forward,
> [34] studied vanilla Q-learning in the tabular MDP setting and
> also adopted Freedman's inequality. However, we would like to argue
> that it requires significant efforts in order to study linear transition
> model and also allow for model misspecification in the current paper.
> In Section C in the supplementary materials, it can be seen that to
> analyze the linear transition model, we encounter a series of technical
> challenges, like how to solve the recursive relationship (53) (whose
> solution is drastically different from [34] as we allow for model
> misspecification) and the establishment of claim (47), which is beyond
> the scope of [34]. Hence, we think the proof techniques, though
> motivated by and has has connections with previous work to some extent,
> has much novelty in the sense that the linearly-parameterized MDP
> requires more sophisticated analysis. We have emphasized the technical
> challenges in our revised paper as follows.
>
> $\mathsf{Revised\ text:} $ "For the model-based approach, we employ
> the leave-one-out analysis to
> decouple the complicated statistical dependency between the empirical
> probability transition model $\widehat{P}$ and the corresponding
> optimal policy. Specifically, [2] proposed to construct a collection
> of auxiliary MDPs where each one of them leaves out a single state
> $s$ by setting $s$ to be an absorbing state and keeping everything
> else untouched. We tailor this high level idea to the needs of linear
> transition model, then the independence between the newly constructed
> MDP with absorbing state $s$ and data samples collected at state
> $s$ will facilitate our analysis, as detailed in Lemma 1. Compared
> with [2], Theorem 1 extends the tabular setting studied in [2]
> to the linear transition model and accommodates model misspecification,
> which actually needs significant efforts as detailed in the supplementary
> materials. This leave-one-out type of analysis has been utilized in studying numerous problems by
> a long line of work, such as [22, 37, 51, 13, 12, 14, 15], just
> to name a few.
>
> To obtain tighter sample complexity bound than the previous one $\widetilde{O}(\frac{K}{(1-\gamma)^{7}\varepsilon^{2}})$
> in [61] for vanilla Q-learning, we invoke Freedman's
> inequality [24] for the concentration of an error term with martingale
> structure as illustrated in Section C in the supplementary materials,
> while the classical ones used in analyzing Q-learning are Hoeffding's
> inequality and Bernstein's inequality [61]. The
> use of Freedman's inequality helps us establish a recursive
> relation on $\\{\Vert Q_{t}-Q^{\star}\Vert_{\infty}\\}_{t=0}^{T}$,
> which consequently leads to the performance guarantee (9). It is worth
> mentioning that [34] also studied vanilla Q-learning in the tabular
> MDP setting and adopted Freedman's inequality, while we emphasize
> that it requires a lot of efforts and more delicate analyses in order
> to study linear transition model and also allow for model misspecification
> in the current paper, as detailed in the supplementary materials.
> "
>
> 3. The $Q$-learning algorithm studied in this paper uses synchronous
> update, while practically $Q$-learning updates $Q_{k}$ in an asynchronous
> manner based a trajectory of samples obtained from a behavioral policy.
>
> $\mathsf{Response:}$ Thank you for your comment! Asynchronous Q-learning
> is indeed a very interesting and practically meaningful problem to
> study. Extending the results in the current paper to asynchronous
> setting call for new algorithmic idea and new analysis tool to deal
> with the Markovian nature of its sampling process, and is one of our
> future research direction.

---

### Official Review · Reviewer_bj21 · 2021-07-19

**Rating:** 7
**Confidence:** 3

**Summary:**

The paper considers reinforcement learning problem in infinite-horizon, discounted reward MDPs with a generative model. The authors aim to obtain upper bounds on the number of samples required to efficiently learn the optimal policy or the optimal $Q$-function when number of states and actions are very large. In order to achieve this, the authors impose generalization by considering transition probabilities to be linear in known state-action features (a.k.a. the linear MDP setting). The authors further assume access to a number of anchor state-action pairs, whose features (essentially) form a basis of the feature space. Under these assumptions on the MDP, the authors propose both model-based and model-free algorithm for learning the optimal policy (or $Q$-function), and analyze their sample complexity guarantees. The authors prove that the sample complexities of both the algorithms scale linearly with the dimension of the feature space (which is also equal to the number of anchor state-action pairs). The sample complexity bound of the model-based algorithm is the first of its kind for linear MDPs, and matches the minimax lower bound of Yang and Wang, 2019. The sample complexity bound of the model-free algorithm, though suboptimal, improves over the previously known bound of Yang and Wang, 2019.

**Limitations And Societal Impact:**

This is a theoretical work, and hence possible implications on the society are not foreseeable.

**Main Review:**

The model based algorithm and its analysis is novel in the context of linear MDPs. The model-free algorithm, though known and analyzed before, obtains an improved sample complexity which is achieved through a novel application of Freedman inequality. However, I have not checked the correctness of the proofs in detail. The results are significant as long as sample complexity bounds in linear MDPS are concerned, but similar results already exist for tabular MDPs. This paper, though provide sample complexity bounds independent of number of state and actions, do not go beyond the tabular setting. The paper is well-written and the flow is clear.

I have one question for the authors. The dimension of the features and the number of anchor state-action pairs are assumed to be equal. What happens and how the results change is these are different?

**Time Spent Reviewing:**

2

---

> ### Author Response · Authors · 2021-08-10
> **Response to Reviewer bj21**
>
> We thank gratefully the reviewer for careful reading of our manuscript
> and for very helpful comments and suggestions. We have tried our best
> to address the comments as follows.
>
> The assumption that the feature dimension (denoted by $K_{\mathsf{d}}$)
> and the number of anchor state-action pairs (denoted by $K_{\mathsf{n}}$)
> are equal is actually non-essential. In what follows, we will show
> that if $K_{\mathsf{d}}\neq K_{\mathsf{n}}$, then we can modify the
> current feature mapping $\phi:\mathcal{S}\times\mathcal{A}\to\mathbb{R}^{K_{\mathsf{d}}}$
> to achieve a new feature mapping $\phi':\mathcal{S}\times\mathcal{A}\to\mathbb{R}^{K_{\mathsf{n}}}$
> that does not change the transition model $P$. By doing so, the new
> feature dimension $K_{\mathsf{n}}$ equals to the number of anchor
> state-action pairs.
>
> To begin with, we recall from Definition 1 that there exists $K_{\mathsf{d}}$
> unknown functions $\psi_{1}$, $\cdots$, $\psi_{K_{\mathsf{d}}}:\mathcal{S}\rightarrow\mathbb{R}$,
> such that
> \begin{equation}
> \qquad\qquad\qquad\qquad\qquad\qquad\qquad\qquad\qquad
> P\left(s'|s,a\right)=\sum_{k=1}^{K_{\mathsf{d}}}\phi_{k}\left(s,a\right)\psi_{k}\left(s'\right),
> \end{equation}
> for every $(s,a)\in\mathcal{S}\times\mathcal{A}$ and $s'\in\mathcal{S}$.
> In addition, we also recall from Assumption 1 that there exists $\mathcal{K}\subseteq\mathcal{S}\times\mathcal{A}$
> with $\vert\mathcal{K}\vert=K_{\mathsf{n}}$ such that for any $(s,a)\in\mathcal{S}\times\mathcal{A}$,
> \begin{equation}
> \qquad\qquad\qquad\qquad
> \phi\left(s,a\right)=\sum_{i:(s_{i},a_{i})\in\mathcal{K}}\lambda_{i}\left(s,a\right)\phi\left(s_{i},a_{i}\right)\in\mathbb{R}^{K_{\mathsf{d}}}\quad\text{for}\quad\sum_{i=1}^{K_{\mathsf{n}}}\lambda_{i}\left(s,a\right)=1\quad\text{and}\quad\lambda_{i}\left(s,a\right)\geq0.
> \end{equation}
>
> $\mathsf{Case\ 1:}$ $K_{\mathsf{d}}>K_{\mathsf{n}}$. In this case, the
> vectors in $\{\phi(s,a):(s,a)\in\mathcal{K}\}$ are linearly independent.
> For ease of presentation and without loss of generality, we assume
> that $K_{\mathsf{d}}=K_{\mathsf{n}}+1$. This indicates that the matrix
> $\Phi\in\mathbb{R}^{K_{\mathsf{d}}\times(\vert\mathcal{S}\vert\vert\mathcal{A}\vert)}$
> whose columns are composed of the feature vectors of all state-action
> pairs has rank $K_{\mathsf{n}}$ and is hence not full row rank. This
> suggests that there exists $K_{\mathsf{n}}$ linearly independent
> rows (without loss of generality, we assume they are the first $K_{\mathsf{n}}$
> rows). We can remove the last row from $\Phi$ to obtain $\Phi':=\Phi_{1:K_{\mathsf{n}},:}\in\mathbb{R}^{K_{\mathsf{n}}\times(\vert\mathcal{S}\vert\vert\mathcal{A}\vert)}$
> such that $\Phi'$ is full row rank. Then we show that we can
> actually use the columns of $\Phi'$ as new feature mappings.
> To see why this is true, note that the last row $\Phi_{K_{\mathsf{n}}+1,:}$
> can be represented as a linear combination of the first $K_{\mathsf{n}}$
> rows, namely there must exist constants $\\{c_{k}:1\\leq k\leq K_{\mathsf{n}}\\}$
> such that for any $(s,a)\in\mathcal{S}\times\mathcal{A}$,
> \begin{equation}
> \qquad\qquad\qquad\qquad\qquad\qquad\qquad\qquad\qquad\qquad
> \phi_{K_{\mathsf{n}}+1}(s,a)=\sum_{k=1}^{K_{\mathsf{n}}}c_{k}\phi_{k}(s,a).
> \end{equation}
> Define $\psi_{k}'=\psi_{k}+c_{k}\psi_{K_{\mathsf{n}}+1}$ for $k=1,\ldots,K_{\mathsf{n}}$,
> we have
> \begin{equation}
> \begin{aligned}
> \qquad\qquad\qquad\qquad\qquad P\left(s'\vert s,a\right) & =\sum_{k=1}^{K_{\mathsf{d}}}\phi_{k}\left(s,a\right)\psi_{k}\left(s'\right)=\phi_{K_{\mathsf{n}}+1}\left(s,a\right)\psi_{K_{\mathsf{n}}+1}\left(s'\right)+\sum_{k=1}^{K_{\mathsf{n}}}\phi_{k}\left(s,a\right)\psi_{k}\left(s'\right)\\\\
>  & =\sum_{k=1}^{K_{\mathsf{n}}}\phi_{k}\left(s,a\right)\left[\psi_{k}\left(s'\right)+c_{k}\psi_{K_{\mathsf{n}}+1}\left(s'\right)\right]=\sum_{k=1}^{K_{\mathsf{n}}}\phi_{k}\left(s,a\right)\psi_{k}'\left(s'\right),
> \end{aligned}
> \end{equation}
> which is linear with respect to the new $K_{\mathsf{n}}$ dimensional
> feature vectors. It is also straightforward to check that the new
> feature mapping satisfies Assumption 1 with the original anchor state-action
> pairs $\mathcal{K}$.
>
> $\mathsf{Case\ 2:}$ $K_{\mathsf{d}}<K_{\mathsf{n}}$. For ease of presentation
> and without loss of generality, we assume that $K_{\mathsf{n}}=K_{\mathsf{d}}+1$
> and that the subspace spanned by the feature vectors of anchor state-action
> pairs is non-degenerate, i.e., has rank $K_{\mathsf{d}}$ (otherwise
> we can use similar method as in Case 1 to further reduce the feature
> dimension $K_{\mathsf{d}}$). In this case, the matrix $\Phi_{\mathcal{K}}\in\mathbb{R}^{K_{\mathsf{d}}\times K_{\mathsf{n}}}$
> whose columns are composed of the feature vectors of anchor state-action
> pairs has rank $K_{\mathsf{d}}$. We can add $K_{\mathsf{n}}-K_{\mathsf{d}}=1$
> new row to $\Phi_{\mathcal{K}}$ to obtain $\Phi_{\mathcal{K}}'\in\mathbb{R}^{K_{\mathsf{n}}\times K_{\mathsf{n}}}$
> such that $\Phi_{\mathcal{K}}'$ has full rank $K_{\mathsf{n}}$.
> Then we let the columns of $\Phi_{\mathcal{K}}'$
> to be the new feature vectors of the anchor state-action pairs $\\{\phi'(s,a):(s,a)\in\mathcal{K}\\}$, and
> define the new feature vectors for all other state-action pairs $(s,a)\notin\mathcal{K}$
> by
> \begin{equation}
> \qquad\qquad\qquad\qquad\qquad\qquad
> \phi'\left(s,a\right)=\sum_{i:(s_{i},a_{i})\in\mathcal{K}}\lambda_{i}\left(s,a\right)\phi'\left(s_{i},a_{i}\right).
> \end{equation}
> We can check that the transition model $P$ is not changed if we let
> $\psi_{K_{\mathsf{n}}}(s')=0$ for every $s'\in\mathcal{S}$. It is
> also straightforward to check that Assumption 1 is satisfied.
>
> To conclude, when $K_{\mathsf{d}}\neq K_{\mathsf{n}}$, we can always
> construct a new set of feature mappings with dimension $K_{\mathsf{n}}$
> such that: (i) the feature dimension equals to the number of anchor
> state-action pairs (they are both $K_{\mathsf{n}}$); (ii) the transition
> model can still be linearly parameterized by this new set of feature
> mappings; and (iii) the anchor state-action pair assumption (Assumption 1) is satisfied with the original anchor state-action pairs. We have
> added a remark in the main text of the revised paper to lead interested
> readers to the above arguments, which we have added in the supplementary
> material.

---

> > ### Comment · Reviewer_bj21 · 2021-08-27
> > **Reply to the rebuttal**
> >
> > I thank the authors for replying to my question, and I will increase my score accrodingly.

---

### Official Review · Reviewer_7gbE · 2021-07-19

**Rating:** 6
**Confidence:** 5

**Summary:**

This paper considers MDPs with linear transition models. Assuming access to a generative model, under certain anchor state assumption, the authors prove the following two results:

1. A model-based approach achieves a sample complexity of $O~(K / ((1 - \gamma)^3 \epsilon^2))$ which matches the minimax optimal lower bound up to logarithm factors;
2. A model-free approach based on Q-learning achieves a sample complexity of  $O~(K / ((1 - \gamma)^3 \epsilon^2))$.

**Limitations And Societal Impact:**

See main review.

**Main Review:**

Understanding the tight sample complexity of RL with linear function approximation is an important problem in RL. This paper improves the state-of-the-art sample complexity for RL with linear function approximation assuming access to a generative model. The authors show that the sample complexity of a model-based approach matches the minimax optimal lower bound up to logarithm factors when $\epsilon$ is sufficiently small. The authors also improve the sample complexity of vanilla Q-learning in this setting.

On the technical side, for the model-based result, the authors borrow the “leave-one-out” analysis developed by Agarwal et al. and generalize it to the linear function approximation setting. For the model-free result, the authors use Freeman's inequality to improve the concentration of the error term.

My major concern is Assumption 1 which is very strong. Although it has been adopted in the RL literature (the authors should consider citing [1] and [2] which also used this assumption) and could be crucial for obtaining tight bounds, the authors should at least discuss whether it is possible to weaken such an assumption. What happens if some $\lambda_i(s, a)$ can be negative but still has bounded absolute value? What happens if we instead assume $\sum \lambda_i(s, a) \le T$ for some $T > 0$? How should the sample complexity depend on $T$? Note that there always exists a set of anchor state-action pairs with $|\lambda_i(s, a)| \le 1$ for all $i \in [K]$ and $(s, a)$ which is implied by the existence of barycentric spanners [3]. This paper could be significantly stronger if such a discussion is available.

Overall, this is an interesting paper, and the writing is easy to follow. However, the main assumption (assumption 1) seems pretty strong, and thus my recommendation is merely a weak acceptance.


[1] Efficient Planning in Large MDPs with Weak Linear Function Approximation

[2] Limiting Extrapolation in Linear Approximate Value Iteration

[3] Online linear optimization and adaptive routing

**Time Spent Reviewing:**

5

---

> ### Author Response · Authors · 2021-08-10
> **Response to Reviewer 7gbE**
>
> We thank gratefully the reviewer for careful reading of our manuscript
> and for very helpful comments and suggestions. We have tried our best
> to address the comments as follows.
>
> 1. We thank the reviewer for suggesting citing [1] and [2],
> which also assume the existence of anchor state-action pairs. We have
> cited both papers in the revised paper.
>
> 2. Thanks for raising this point! While we agree that Assumption 1
> is quite strong, we believe that within the current analysis framework,
> it is very difficult to obtain tight sample complexity bounds without
> assuming $\lambda_{i}(s,a)\geq0$ and $\sum_{i=1}^{K}\lambda_{i}(s,a)=1$.
> First we note that equation (3) in the current paper, i.e.
> \begin{equation}
> \qquad\qquad\qquad\qquad\qquad\qquad\qquad
> P\left(\cdot\vert s,a\right)=\sum_{i:(s_{i},a_{i})\in\mathcal{K}}\lambda_{i}\left(s,a\right)P\left(\cdot\vert s_{i},a_{i}\right)
> \end{equation}
> holds even without assuming $\lambda_{i}(s,a)\geq0$ and $\sum_{i=1}^{K}\lambda_{i}(s,a)=1$.
> In order to guarantee that $P(\cdot\vert s,a)$ on the left hand side
> is a probability distribution, it is necessary and inevitable to assume
> that $\sum_{i=1}^{K}\lambda_{i}(s,a)=1$ (note that the state-action
> pairs constructed using [3] does not necessarily satisfy $\sum_{i=1}^{K}\lambda_{i}(s,a)=1$).
> Therefore the restriction imposed on $\\{\lambda_{i}(s,a):1\leq i\leq K\\}$ by Assumption 1 is basically $\lambda_i(s,a)\geq0$, which is
> equivalent to $\sum\vert\lambda_{i}(s,a)\vert=1$.
> We think it is difficult to
> modify our proof technique to allow some $\lambda_i(s,a)$ to be
> negative. An attempt in this direction was made in prior work [4],
> which only assumed $\sum\vert\lambda_{i}(s,a)\vert\leq L$ for some
> $L\geq1$ and allowed some $\lambda_{i}(s,a)$ to be negative. However
> Theorem 2 therein proved that the required sample complexity for their
> Phased Parametric Q-learning to learn an $\varepsilon$-optimal policy
> is $\widetilde{O}(\frac{KL^{2}}{(1-\gamma)^{7}\varepsilon^{2}})$.
> The dependency on the effective horizon in their bound, i.e., $(1-\gamma)^{-7}$,
> is not tight according to our discussion after Theorem 2 in the current
> paper. We have added the following discussion in our revised paper.
>
> $\mathsf{Revised\ text:}$ "However it is worth mentioning that [61,
> Theorem 2] is built upon weaker conditions $\sum_{i=1}^{K}\lambda_{i}(s,a)=1$
> and $\sum_{i=1}^{K}\vert\lambda_{i}(s,a)\vert\leq L$ for some $L\geq1$,
> which does not require $\lambda_{i}(s,a)\geq0$. Our result holds
> under Assumption 1, which requires $\sum_{i=1}^{K}\lambda_{i}(s,a)=1$
> and $\lambda_{i}(s,a)\geq0$. Under the current analysis framework,
> it is difficult to obtain tight sample complexity bounds without assuming
> $\lambda_{i}(s,a)\geq0$.''
>
> References:
>
> [1] Shariff, Roshan, and Csaba Szepesvári. Efficient
> planning in large MDPs with weak linear function approximation.
> arXiv preprint arXiv:2007.06184 (2020).
>
> [2] Zanette, Andrea, Alessandro Lazaric, Mykel J. Kochenderfer,
> and Emma Brunskill. Limiting Extrapolation in Linear
> Approximate Value Iteration. Advances in Neural Information
> Processing Systems 32 (2019): 5615-5624.
>
> [3] Awerbuch, Baruch, and Robert Kleinberg. Online
> linear optimization and adaptive routing. Journal of
> Computer and System Sciences 74, no. 1 (2008): 97-114.
>
> [4] Yang, Lin, and Mengdi Wang. Sample-optimal
> parametric Q-learning using linearly additive features.
> In International Conference on Machine Learning, pp. 6995-7004. PMLR,
> 2019.

---

### Decision · Program_Chairs · 2021-09-27

**Decision:**

Accept (Poster)

**Comment:**

The paper has initially received mixed reviews, with several reviewers taking issue with the strength of the assumptions. After reading the very detailed author response and some further discussion, two reviewers have decided to raise their scores, so eventually all scores ended up being positive. The reviewers agreed that the paper offers an interesting technical contribution clearly improving the previous state of the art, and that it is worthy of being published at the conference. Based on my own reading, I concur with this assessment and strongly recommend this paper for acceptance.